# VULNERABILITY-AWARE POISONING MECHANISM FOR ONLINE RL WITH UNKNOWN DYNAMICS

**Yanchao Sun**[1]      **Da Huo**[2]      **Furong Huang**[3]

[1,3] Department of Computer Science, University of Maryland, College Park, MD 20742, USA
[2] Shanghai Jiao Tong University, China
[1]`ycs@umd.edu`, [2]`sjtuhuoda@sjtu.edu.cn`, [3]`furongh@umd.edu`

## ABSTRACT

Poisoning attacks on Reinforcement Learning (RL) systems could take advantage of RL algorithm's vulnerabilities and cause failure of the learning. However, prior works on poisoning RL usually either unrealistically assume the attacker knows the underlying Markov Decision Process (MDP), or directly apply the poisoning methods in supervised learning to RL. In this work, we build a generic poisoning framework for online RL via a comprehensive investigation of heterogeneous poisoning models in RL. Without any prior knowledge of the MDP, we propose a strategic poisoning algorithm called Vulnerability-Aware Adversarial Critic Poison (VA2C-P), which works for on-policy deep RL agents, closing the gap that no poisoning method exists for policy-based RL agents. VA2C-P uses a novel metric, stability radius in RL, that measures the vulnerability of RL algorithms. Experiments on multiple deep RL agents and multiple environments show that our poisoning algorithm successfully prevents agents from learning a good policy or teaches the agents to converge to a target policy, with a limited attacking budget.

## 1 INTRODUCTION

Although reinforcement learning (RL), especially deep RL, has been successfully applied in various fields, the security of RL techniques against adversarial attacks is not yet well understood. In real-world scenarios, including high-stakes ones such as autonomous driving vehicles and healthcare systems, a bad decision may lead to a tragic outcome. Should we trust the decision made by an RL agent? How easy is it for an adversary to mislead the agent? These questions are crucial to ask before deploying RL techniques in many applications.

In this paper, we focus on *poisoning attacks*, which occur during the training and influence the learned policy. Since training RL is known to be very sample-consuming, one might have to constantly interact with the environment to collect data, which opens up a lot of opportunities for an attacker to poison the training samples collected. Therefore, understanding poisoning mechanisms and studying the vulnerabilities in RL are crucial to provide guidance for defense methods. However, existing works on adversarial attacks in RL mainly study the test-time *evasion attacks* (Chen et al., 2019) where the attacker crafts adversarial inputs to fool a well-trained policy, but does not cause any change to the policy itself. Motivated by the importance of understanding RL security in the training process and the scarcity of relevant literature, in this paper, we *investigate how to poison RL agents and how to characterize the vulnerability of deep RL algorithms*.

In general, RL is an "online" process: an agent rolls out experience from the environment with its current policy, and uses the experience to improve its policy, then uses the new policy to roll out new experience, etc. Poisoning in online RL is significantly different from poisoning in classic supervised learning (SL), even online SL, and is more difficult due to the following challenges.

*Challenge I – Future Data Unavailable in Online RL.* Poisoning approaches in SL (Muñoz-González et al., 2017; Wang & Chaudhuri, 2018) usually require the access to the whole training dataset, so the attacker can decide the optimal poisoning strategy before the learning starts. However, in online RL, the training data (trajectories) are generated by the agent while it is learning. Although the optimal poison should work in the long run, the attacker can only access and change the data in the current iteration, since the future data is not yet generated.

*Challenge II – Data Samples No Longer i.i.d..* It is well-known that in RL, data samples (state-action transitions) are no longer i.i.d., which makes learning challenging, since we should consider the long-term reward rather than the immediate result. How-

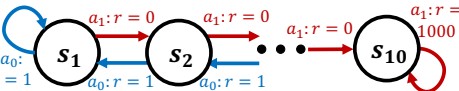

**Figure 1:** An example of difficult poisoning.

ever, we notice that data samples being not i.i.d. also makes poisoning attacks challenging. For example, an attacker wants to reduce the agent's total reward in a task shown as Figure 1; at state $s_1$, the attacker finds that $a_1$ is less rewarding than $a_0$; if the attacker only looks at the immediate reward, he will lure the agent into choosing $a_1$. However, following $a_1$ finally leads the agent to $s_{10}$ which has a much higher reward.

*Challenge III – Unknown Dynamics of Environment.* Although Challenge I and II can be partially addressed by predicting the future trajectories or steps, it requires prior knowledge on the dynamics of the underlying MDP. Many existing poisoning RL works (Rakhsha et al., 2020; Ma et al., 2019) assume the attacker has perfect knowledge of the MDP, then compute the optimal poisoning. However, in many real-world environments, knowing the dynamics of the MDP is difficult. Although the attacker could potentially interact with the environment to build an estimate of the environment model, the cost of interacting with the environment could be unrealistically high, market making (Spooner et al., 2018) for instance. In this paper, we study a more realistic scenario where the attacker does not know the underlying dynamics of MDP, and can not directly interact with the environment, either. Thus, the attacker learns the environment only based on the agent's experience.

In this paper, we systematically investigate poisoning in RL by considering all the aforementioned RL-specific challenges. Previous works either do not address any of the challenges or only address some of them. Behzadan & Munir (2017) achieve policy induction attacks for deep Q networks (DQN). However, they treat output actions of DQN similarly to labels in SL, and do not consider Challenge II that the current action will influence future interactions. Ma et al. (2019) propose a poisoning attack for model-based RL, but they suppose the agent learns from a batch of given data, not considering Challenge I. Rakhsha et al. (2020) study poisoning for online RL, but they require perfect knowledge of the MDP dynamics, which is unrealistic as stated in Challenge III.

**Summary of Contributions. (1)** We propose a practical poisoning algorithm called Vulnerability-Aware Adversarial Critic Poison (VA2C-P) that works for deep policy gradient learners without any prior knowledge of the environment. To the best of our knowledge, VA2C-P is *the first practical algorithm that poisons policy-based deep RL methods.* **(2)** We introduce a novel metric, called stability radius, to characterize the stability of RL algorithms, measuring and comparing the vulnerabilities of RL algorithms in different scenarios. **(3)** We conduct a series of experiments for various environments and state-of-the-art deep policy-based RL algorithms, which demonstrates RL agents' vulnerabilities to even weaker attackers with limited knowledge and attack budget.

## 2 RELATED WORK

The main focus of this paper is on poisoning RL, an emerging area in the past few years. We survey related works of adversarial attacks in SL and evasion attacks in RL in Appendix A, as they are out of the scope of this paper.

**Targeted Poisoning Attacks for RL.** Most RL poisoning researches work on targeted poisoning, also called policy teaching, where the attacker leads the agent to learn a pre-defined target policy. Policy teaching can be achieved by manipulating the rewards (Zhang & Parkes, 2008; Zhang et al., 2009) or dynamics (Rakhsha et al., 2020) of the MDP. However, they require the attackers to not only have prior knowledge of the environments (e.g., the dynamics of the MDP), but also have the ability to *alter the environment* (e.g. change the transition probabilities), which are often unrealistic or difficult in practice.

**Poisoning RL with Omniscient Attackers.** Most guaranteed poisoning RL literature (Rakhsha et al., 2020; Ma et al., 2019) assume *omniscient attackers*, who not only know the learner's model, but also know the underlying MDP. However, as motivated in the introduction, the underlying MDP is usually either unknown or too complex in practice. Some works poison RL learners by changing the reward signals sent from the environment to the agent. For example, Ma et al. (2019) introduce a policy teaching framework for batch-learning model-based agents; Huang & Zhu (2019) propose a reward-poisoning attack model, and provide convergence analysis for Q-learning; Zhang et al.

(2020b) present an adaptive reward poisoning method for Q-learning (while it also extends to DQN) and analyze the safety thresholds of RL; these papers all assume the attacker knows not only the models of the agent, but also the parameters of the underlying MDP, which could be possible in a tabular MDP, but hard to realize in large environments and modern deep RL systems.

On the contrary, *we consider non-omniscient attackers* who do not know the underlying MDP or environment in this paper. The non-omniscient attackers can be further divided into two categories: *white-box* attackers, who know the learner's model/parameters, and *black-box* attackers, who do not know the learner's model/parameters. They both tap the interactions between the learner and the environment.

**Black-box Poisoning for Value-based Learners.** Although there are many successful black-box evasion approaches (Xinghua et al., 2020; Inkawhich et al., 2020), black-box poisoning in RL is rare. There is a *black-box* attacking method for a value-based learner (DQN) proposed by Behzadan & Munir (2017), which does not require the attacker to know the learner's model or the underlying MDP. In this work, the attacker induces the DQN agent to output the target action by perturbing the state with Fast Gradient Sign Method (FGSM) (Goodfellow et al., 2015) in every step. However, the data-correlation problem of RL (Challenge II) is not considered, and FGSM attack does not work for policy-based methods due to their high stochasticity, as we show in experiments.

In this paper, we propose a new poisoning algorithm for *policy-based deep RL* agents, which can achieve *both non-targeted and targeted* attacks. We do not require any prior knowledge of the environment. And our algorithm works not only when the attacker knows the learner's model (white-box), but also when the leaner's model is hidden (black-box).

## 3 PROBLEM FORMULATION FOR POISONING ONLINE RL

### 3.1 NOTATIONS AND PRELIMINARIES

In RL, an agent interacts with the environment by taking actions, observing states and receiving rewards. The environment is modeled by a Markov Decision Process (MDP), which is denoted by a tuple $\mathcal{M} = \langle \mathcal{S}, \mathcal{A}, P, R, \gamma, \mu \rangle$, where $\mathcal{S}$ is the state space, $\mathcal{A}$ is the action space, $P$ is the transition kernel, $R$ is the reward function, $\gamma \in (0, 1)$ is the discount factor, and $\mu$ is the initial state distribution. A *trajectory* $\tau \sim \pi$ generated by *policy* $\pi$ is a sequence $s_1, a_1, r_1, s_2, a_2, \cdots$, where $s_1 \sim \mu$, $a_t \sim \pi(a|s_t)$, $s_{t+1} \sim P(s|s_t, a_t)$ and $r_t = R(s_t, a_t)$. The goal of an RL agent is to find an optimal policy $\pi^*$ that maximizes the *expected total rewards* $\eta$, which is defined as $\eta(\pi) = \mathbb{E}_{\tau \sim \pi}[r(\tau)] = \mathbb{E}_{s_1, a_1, \cdots \sim \mu, \pi, P, R}[\sum_{t=1}^{\infty} \gamma^{t-1} r_t]$.

We use an overhead check sign ˘ on a variable to denote that the variable is poisoned. For example, if the attacker perturbs a reward $r_t$, then the poisoned reward is denoted as $\check{r}_t$. If a policy $\pi$ is updated with poisoned observation, then it is denoted as $\check{\pi}$.

### 3.2 THE PROCEDURE OF ONLINE LEARNING AND POISONING

**Procedure of Online Learning.** We consider a classical online policy-based RL setting, where the learner iteratively updates its *policy* $\pi$ parametrized by $\theta$, through $K$ iterations with the environment. For notation simplicity, we omit $\theta$ and use $\pi_k$ to denote $\pi_{\theta_k}$, the learner's policy at iteration $k$. The online learning process is described as below.

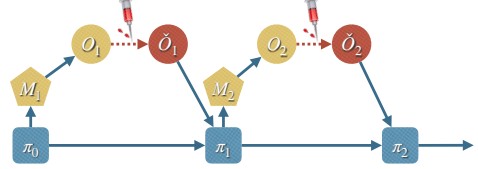

**Figure 2:** Online poisoning-learning process.

At iteration $k = 1, \cdots, K$,
**(1)** The agent uses the current policy $\pi_k$ to roll out **observation** $\mathcal{O}_k = (\mathcal{O}_k^s, \mathcal{O}_k^a, \mathcal{O}_k^r)$ from environment $\mathcal{M}_k$, where $\mathcal{O}_k^s = [s_1, s_2, \cdots], \mathcal{O}_k^a = [a_1, a_2, \cdots], \mathcal{O}_k^r = [r_1, r_2, \cdots]$ are respectively the sequence of states, actions and rewards generated at iteration $k$.
**(2)** The agent updates its policy parameters $\theta$ with its algorithm $f$. Most policy-based algorithms perform *on-policy* updating, i.e., update policy only by the current observation $\mathcal{O}_k$. The on-policy update can be then formalized as $\pi_{k+1} = f(\pi_k, \mathcal{O}_k) \approx \operatorname{argmax}_\pi J(\pi, \pi_k, \mathcal{O}_k)$, where $J$ is an objective function defined by algorithm $f$, e.g., the expected total reward $\eta(\pi)$.

**Procedure of Online Poisoning.** A poisoning attacker influences the learner in the training process by perturbing the training data. In SL, the training data consists of features and labels, and the attacker poisons the training data before the learning starts. However, the training data in RL is the trajectories a learner rolls out from the environment, i.e., observation $\mathcal{O} = (\mathcal{O}^s, \mathcal{O}^a, \mathcal{O}^r)$. At iteration $k$, the attacker eavesdrops on the interaction between the learner and the environment, obtains the observation $\mathcal{O}_k$, and may poison it into $\check{\mathcal{O}}_k$, then send $\check{\mathcal{O}}_k$ to the learner before policy updating. [1] Procedure 2 in Appendix B illustrates how the online game goes between the learner and the attacker. Figure 2 visualizes this online learning-poisoning procedure, where we can see that learning and poisoning are convoluted and inter-dependent.

### 3.3 A Unified Formulation for Poisoning Online Policy-based RL.

#### 3.3.1 Attacker's Poison Aim

As defined in Section 3.1, the observation is a collection of trajectories, consisting of the observed states $\mathcal{O}^s$, the executed actions $\mathcal{O}^a$ or the received rewards $\mathcal{O}^r$. When poisoning the observation, the attacker may only focus on the states, or on the actions, or on the rewards. We call the quantity being altered as the **poison aim** of the attacker, denoted by $\mathcal{D} \in \{\mathcal{O}^s, \mathcal{O}^a, \mathcal{O}^r\}$. For example, $\mathcal{D} = \mathcal{O}^s$ means the attacker chooses to attack the states.

Distinguishing different poison aims is important, since in real-world applications different poison aims correspond to different behaviors of the attacker. For example, in an RL-based recommender system, the RL agent recommends an item (i.e., an action) for a user (i.e., a state), and the user may or may not choose to click on the recommended item (i.e., a reward). An adversary might manipulate the reward (poisoning $\mathcal{D} = \mathcal{O}^r$), e.g., blocking the user's click from the RL agent or creating a fake click. An adversary might also alter the state (poisoning $\mathcal{D} = \mathcal{O}^s$), e.g., raising a teenager user's age which could result in inappropriate recommendations. An adversary might also change the action (poisoning $\mathcal{D} = \mathcal{O}^a$), e.g., inserting a fake recommendation into the agent's list of recommendations. Under different scenarios, the feasibility of poisoning different aims may vary.

Most existing works on poisoning RL only solve one type of poison aim. Zhang et al. (2020b); Huang & Zhu (2019) propose to poison rewards, and Behzdan & Munir (2017) assume the attacker poison the states. However, in our paper, we provide a general solution for any of the poison aims to satisfy the needs in different scenarios. Our proposed method also supports a "hybrid" poison aim, where the attacker could switch aims at different iterations, as discussed in Section 5.

#### 3.3.2 A Poisoning Framework for RL

We focus on proposing a poisoning mechanism for the above challenging online learning scenario. We first formalize the poisoning attacking at iteration $k$ as a *sequential bilevel optimization* problem in Problem (Q), and explain the details of the problem in the remaining of this section.

$$\underset{\check{\mathcal{D}}_k, \cdots, \check{\mathcal{D}}_K}{\text{argmin}} \quad \sum_{j=k}^{K} \lambda_j L_A(\check{\pi}_{j+1}) \qquad \qquad \text{((a) attacker's weighted loss)} \quad \text{(Q)}$$

$$s.t. \qquad \check{\pi}_{j+1} = \text{argmax}_\pi J(\pi, \check{\pi}_j, \check{\mathcal{O}}_j | \check{\mathcal{D}}_j), \forall k \le j \le K \qquad \text{((b) imitate the learner)}$$

$$\sum_{j=1}^{K} \mathbf{1}\{\check{\mathcal{D}}_j \ne \mathcal{D}_j\} \le C \qquad \qquad \text{((c) limited-budget)}$$

$$U(\mathcal{D}_j, \check{\mathcal{D}}_j) \le \epsilon, \forall 1 \le j \le K \qquad \qquad \text{((d) limited-power)}$$

*(a) Attacker's Weighted Loss.* $L_A(\check{\pi})$ measures the attacker's loss w.r.t. a poisoned policy $\pi$. As the definition of poisoning implies, the attacker influences or misleads the learner's policy. $\lambda_{k:K}$ are the weights of future attacker losses, controlling how much the attacker value the poisoning results in different iterations. The goal of the attacker is either (1) **non-targeted poisoning**, which minimizes the expected total rewards of the learner, i.e., $L_A = \eta(\check{\pi})$, or (2) **targeted poisoning**, which induces the learner to learn a pre-defined target policy, i.e., $L_A = \text{distance}(\check{\pi}, \pi^\dagger)$, where $\text{distance}(\check{\pi}, \pi^\dagger)$ can be any distance measure between a learned policy $\check{\pi}$ and a target policy $\pi^\dagger$. Note that the targeted

---

[1] In this paper, we assume the attacker poisons the observation(trajectories), which is the most universal setting in practice. Appendix B extends our problem formulation to a more general case where the attacker can change the underlying MDP.

poisoning objective can usually be directly computed with a prior target policy, while non-targeted poisoning has a "reward-minimizing" objective, which is the reverse of the learner's objective. Without any prior knowledge of the environment, non-targeted poisoning is usually more difficult, as the attacker needs to first learn "what is the worst way" (which is as difficult as a learning problem by a RL agent) and then lead the learner to that way (which is as difficult as a targeted poisoning problem, assuming leading an agent to different policies is roughly equally challenging). However, most existing poisoning work focus on targeted poisoning, which requires the attacker to know a pre-defined target policy. Thus in this paper, we make more efforts to solve the reward-minimizing poisoning problem, which may deprave the policy without any prior knowledge.

*(b) Imitate the Policy-Based Learner.* To confidently mislead a learner, the attacker needs to predict how the learner will behave under the poison, which can be achieved by *imitating the learner using the learner's observation*. More specifically, at the $j$-th iteration, the attacker estimates the learner's policy to be $\tilde{\pi}_j$, called **imitating policy**. Then, the attacker predicts the next-policy $\check{\pi}_{j+1}$ under poisoned observation, based on the learner's update rule $\mathrm{argmax}_\pi J(\pi, \tilde{\pi}_j, \check{\mathcal{O}}_j | \check{\mathcal{D}}_j)$, where $\mathcal{D} \in \{\mathcal{O}^s, \mathcal{O}^a, \mathcal{O}^r\}$ stands for the poison aim of the poisoning, $\check{\mathcal{O}}|\check{\mathcal{D}}$ denotes that $\mathcal{O}$ is poisoned into $\check{\mathcal{O}}$ given that poison aim $\mathcal{D}$ is poisoned into $\check{\mathcal{D}}$. However, the imitating policy $\tilde{\pi}$ may or may not be the same as the actual learner's policy, depending on the *attacker's knowledge*. As introduced in Section 2, we deal with both white-box and black-box attackers, and both of them do not know the environment $\mathcal{M}$. A **white-box** attacker knows the current and past observations $\mathcal{O}_{1:k}$, the learner's algorithm $f$ and policy $\pi$, so it can directly copy the policy $\tilde{\pi}_j = \pi_j, \forall j$. A **black-box** attacker knows the current and past observations $\mathcal{O}_{1:k}$, but does not know the learner's policy $\pi$. In this case, the attacker has to estimate $\pi$ at every iteration. Section 4.3 states how to guess $\pi$.

*(c,d) Limited-budget and Limited-power.* In practice, the ability of an attacker is usually restricted by some constraints. For the online poisoning problem, we consider attacker's constraints in two forms: (1) (**attack budget** $C$) the total number of iterations that the attacker could poison does not exceed $C$; (2) (**attack power** $\epsilon$) in one iteration, the total change[2] $U(\mathcal{D}_k, \check{\mathcal{D}}_k)$ between $\mathcal{D}_k$ and $\check{\mathcal{D}}_k$ can not be larger than $\epsilon$. Attack power controls the amount of perturbation, as commonly used in the adversarial learning literature. Attack budget considers the frequency of attack, which is similar to the constraint studied by Wang & Chaudhuri (2018).

Problem (Q) is a generic formulation, covering a variety of poisoning models, and specifies the best poison an attacker can execute. However, directly solving Problem (Q) is prohibitive, as (1) the future observations $\mathcal{O}_{k+1:K}$ are unknown when poisoning the $k$-th iteration, as the attacker has no knowledge of the underlying MDP. (2) the limited-budget constraint is analogous to $\ell_0$-norm regularization, which is generally NP-hard (Nguyen et al., 2019); and (3) minimizing attacker's loss while maximizing learner's gain is non-convex minimax optimization, which is a complex problem (Perolat et al., 2015).

In spite of the above difficulties, we introduce a practical method to approximately and effectively solve Problem (Q) in Section 4.

## 4 VA2C-P: POISON POLICY GRADIENT LEARNERS

In this section, we propose a practical and efficient poisoning algorithm called Vulnerability-Aware Adversarial Critic Poison (VA2C-P) for policy gradient learners. Without loss of generality, we assume the loss weights $\lambda_j = 1$ for all $j = 1, \cdots, K$.

**Main Idea.** As discussed in Section 3.3, Problem (Q) is difficult mainly because of the unknown future observations and the limited budget constraint. In other words, it is hard to exactly determine (1) what kind of attack benefits the future the most, and (2) which iterations are worth attacking the most. Thus, we propose to break Problem (Q) into two decisions: when to attack, and how to attack. The "when to attack" decision allocates the limited budget to iterations which are more likely to be influenced by the attacker, and the "how to attack" decision utilizes limited power to minimize the attacker's loss. We introduce two mechanisms of VA2C-P, vulnerability-awareness and adversarial critic, to make these two decisions respectively.

---

[2]There are many choices of $U(\cdot, \cdot)$. For example, the total effort w.r.t. $\mathcal{O}^s$-poisoning can be the average $\ell_p$-distance between any poisoned and unpoisoned state in $\mathcal{O}^s$ and $\check{\mathcal{O}}^s$.

### 4.1 DECISION 1: WHEN TO ATTACK – VULNERABILITY-AWARE

To answer the question of when to attack, we identify the iterations under which the learner's policy gets more depraved by the same level of attack power. Inspired by the notion of stability in learning theory, which measures how a machine learning algorithm changes due to a small perturbation of the input data, we formally investigate the stability of an RL algorithm, which is the first attempt in the existing literature to the best of our knowledge.

**Stability of RL Algorithms.** We first focus on a single update process of an algorithm $f$. An update $\pi' = f(\pi, \mathcal{O})$ is stable if a limited-power poisoning attack does not cause any difference on the output policy $\pi'$. That is, the learning algorithm produces the same result regardless of the presence of the poison. More formally, we define the concept of *stability radius of one update* in Definition 1.

**Definition 1** (Stability Radius of One Update). [3] *For the update of an RL algorithm $\pi' = f(\pi, \mathcal{O})$, with any poison aim $\mathcal{D}$, the $\delta$-stability radius of the update is defined as the minimum poison power needed to cause $\delta$ change in policy (called $\delta$-policy-discrepancy)*

$$\phi_{\delta,\mathcal{D}}(f, \pi, \mathcal{O}) = \inf_{\epsilon}\{\exists \check{\mathcal{D}} \ s.t. \ U(\mathcal{D}, \check{\mathcal{D}}) \leq \epsilon \ \text{ and } \ d^{\max}[\pi'||\check{\pi}'] > \delta, \ \text{ where } \ \check{\pi}' = f(\pi, \check{\mathcal{O}}|\check{\mathcal{D}})\}, \quad (1)$$

*Policy discrepancy $d^{\max}[\pi_1||\pi_2] = \max_s d\big[\pi_1(\cdot|s)||\pi_2(\cdot|s)\big]$, where $d[\cdot||\cdot]$ could be any measure of distribution distance.*

**Remarks.** (1) The one-update stability radius is w.r.t. the algorithm $f$, the old policy $\pi$, the clean observation $\mathcal{O}$ and the poison aim $\mathcal{D}$. (2) Poison with power under $\phi_{\delta,\mathcal{D}}(f, \pi, \mathcal{O})$ will not cause the policy distributions to change more than $\delta$. (3) As shown by Proposition 2 in Appendix D.1, poison with power under $\phi_{\delta,\mathcal{D}}(f, \pi, \mathcal{O})$ will not make the policy value drop more than $O\big(\delta^2\gamma(1 - \gamma)^{-2}\max_{s,a}|A_{\pi'}(s, a)|\big)$, where $A$ is the advantage function, i.e., $A_\pi(s, a) = Q_\pi(s, a) - V_\pi(s)$.

Stability radius measures the minimal effort needed to make the *poisoned next-policy* $\check{\pi}'$ notably different from the *clean next-policy* $\pi'$ which the learner will get if no poison is applied. Assuming $\max_{s,a}|A_{\pi'}(s, a)|$ does not drastically change for the learner's policy during training, then an attack that causes higher policy discrepancy between $\pi'$ and $\check{\pi}$ could cause more drop of the policy value. Therefore, the idea of vulnerability-aware attack is to estimate the vulnerability of each update and attack the most vulnerable ones. More specifically, if the attacker finds an $\epsilon$-powered attack results in a policy discrepancy larger than some threshold $\delta$, then it can conclude $\phi_\delta(f, \pi, \mathcal{O}) \leq \epsilon$, and the current update is relatively vulnerable. In practice, we have the fixed budget $C$ instead of $\delta$, so we perform the vulnerability check in an adaptive way: attacking the $C$ iterations where $\epsilon$-powered attacks can trigger the highest policy discrepancies. Section 4.3 introduces an algorithm to realize this idea.

### 4.2 DECISION 2: HOW TO ATTACK – ADVERSARIAL CRITIC

Since "when to attack" decision tackles the limited-budget constraint, now we turn our attention to minimizing the attacker's loss while satisfying the limited-power constraint. If the attacker has already decided to poison the $k$-th iteration due to its high vulnerability, then the decision of how to attack should be made before the learner uses the observation to update the policy. We relax the original Problem (Q) into Problem (P) as below.

$$\text{argmin}_{\check{\mathcal{D}}_k} \quad L_A(\check{\pi}_{k+1}) \quad\quad\quad\quad\quad\quad\quad\quad\quad\quad (\text{P})$$
$$s.t. \quad\quad \check{\pi}_{k+1} = \text{argmax}_\pi J(\pi, \check{\pi}_k, \check{\mathcal{O}}_k|\check{\mathcal{D}}_k)$$
$$U(\mathcal{D}_k, \check{\mathcal{D}}_k) \leq \epsilon$$

Compared with Problem (Q), Problem (P) does not consider future losses, which require the unavailable future observations. Instead, Problem (P) finds a greedy attack $\check{\mathcal{D}}_k$ to minimize the loss of the immediate next iteration. The solution to Problem (P) is always feasible to Problem (Q), although might not be optimal. Appendix F.2 provides details about the optimality of Problem (Q).

In practice, it is also challenging to estimate $L_A(\check{\pi})$. For targeted attacking, $L_A = \text{distance}(\check{\pi}, \pi^\dagger)$ is directly computable with a properly defined distance metric, but for non-targeted attacking, the loss

---

[3] In order to characterize the stability and robustness of RL algorithms in a principled way, we provide more measures for the vulnerability of RL in both training time and test time. See Appendix D.2) for details.

$L_A = \eta(\check{\pi})$ can not be directly computed, since $\check{\pi}$ is not the behavior policy. Although one can use importance sampling to evaluate $\eta(\check{\pi})$ with the current trajectories generated by the learner's policy, i.e., $\mathbb{E}_{\tau \sim \pi_{k-1}}[\frac{\pi_k(\tau)}{\pi_{k-1}(\tau)}r(\tau)]$, it may suffer from a high variance (Schulman et al., 2015b) when there are few trajectories. To solve this challenge, we introduce another mechanism, adversarial critic.

**Adversarial Critic.** Under poisoning, the value network (if any) held by the learner usually fails to fit the correct value of its policy, since it does not observe the real trajectories generated by its policy. However, the attacker observes the real trajectories before poisoning, and is able to learn the real values to make the attacking stronger. Inspired by the Actor-Critic method, we propose to let the attacker learn a value function (network) $\tilde{V}_\omega$ with observations of the learner, i.e., the attacker learns a critic of the learner's current policy $\pi$. Then the attacker can use $\tilde{V}_\omega$ to design poisoned observations, directing the learner to a decreasing-value direction, which is called *Adversarial Critic*. Then, using importance sampling, the attacker's loss becomes $\mathbb{E}_{s,a \sim \pi_k}[\frac{\tilde{\pi}(a|s)}{\pi_k(a|s)}(G(s_t, a_t) - \tilde{V}_\omega(s_t))]$, where $G$ is the discounted future reward $\sum_{i=t}^{T} \gamma^{i-t} r_t$.

### 4.3 Poisoning Algorithm VA2C-P.

---

**Algorithm 1:** Vulnerability-Aware Adversarial Critic Poison

---

**Input:** total iterations of learning $K$; poisoning power $\epsilon$; poisoning budget $C$;

1   Initialize a list of policy discrepancies $\Psi = \emptyset$, the number of poisoned iterations $c = 0$
2   Initialize an adversarial critic network $\tilde{V}_\omega$ and an imitating policy network $\tilde{\pi}$
3   **for** $k = 1, \cdots, K$ **do**
4      **if** $c > C$ **then**
5         Break
6      Obtain the observation (trajectories) $\mathcal{O}_k$ obtained by the learner
7      Update the adversarial critic $\tilde{V}_\omega$ with observation $\mathcal{O}_k$
8      **if** *White-box* **then**
9         Copy the learner's policy parameters to the imitating policy of attacker: $\tilde{\pi} \leftarrow \pi_k$
10      Compute the clean next-policy $\pi' \leftarrow f(\tilde{\pi}, \mathcal{O}_k)$
11      Solve Problem (P) with power $\epsilon$ and critic $\tilde{V}_\omega$, poison $\mathcal{O}_k$ to $\check{\mathcal{O}}_k$
12      Compute the poisoned next-policy $\check{\pi}' \leftarrow f(\tilde{\pi}, \check{\mathcal{O}}_k)$
13      Estimate the policy discrepancy $\widehat{\psi}_k$ between $\pi'$ and $\check{\pi}'$
14      Add $\widehat{\psi}_k$ to the list of policy discrepancies $\Psi$
15      **if** $\widehat{\psi}_k \geq \lfloor \frac{C-c}{K-k} \rfloor$*-th largest element in* $\Psi$ **then**
16         Send the learner the poisoned observation $\check{\mathcal{O}}_k$
17         **if** *Black-box* **then**
18            Update the imitating policy as the clean next-policy: $\tilde{\pi} \leftarrow \pi'$
19      **else**
20         Send the clean observation $\mathcal{O}_k$
21         **if** *Black-box* **then**
22            Update the imitating policy as the poisoned next-policy: $\tilde{\pi} \leftarrow \check{\pi}'$

---

Algorithm 1 illustrates how an attacker can poison an online RL learner with our proposed VA2C-P, which is corresponding to Line 4 in Procedure 2 shown in Appendix B . More implementation details are in Algorithm 3 in Appendix E.

In an iteration, an attacker first finds a good perturbation for observation $\mathcal{O}$ with the current attack power $\epsilon$, then estimates how much the output policy will change by computing the policy discrepancy between the clean next-policy and the poisoned next-policy. Then the attacker poisons if the policy discrepancy ranks high in the historical policy discrepancies. In Line 11, we use projected gradient descent to solve Problem P. Computation details are illustrated in Appendix E.

Algorithm 1 covers both white-box and black-box settings, both of which maintain an imitating policy $\tilde{\pi}$ to keep track of the learner's potential status. As mentioned in Section 3.3, a white-box attacker knows the learner's current policy, thus he can directly copy the learner's parameters to

his imitating policy (Line 9), then predict the next-policy the learner would get under different observations. In contrast, a black-box attacker does not know the learner's policy, but he can update its imitating policy using the same observation as the learner uses (Line 18, 22). As claimed and verified by many black-box attacking methods (Behzadan & Munir, 2017), adversarial attacks are usually transferable, i.e., if the attack works on the imitating learner, then the attack is also likely to work for the real learner. Note that as assumed by Behzadan & Munir (2017), the black-box attacker knows what RL algorithm the learner is using (e.g., PPO, A2C, etc), so that the black-box attacker computes its estimation for the next-policy in Line 10 and Line 12.

## 5 EXPERIMENTS

In this section, we evaluate the performance of VA2C-P by poisoning multiple algorithms on various environments. We demonstrate that VA2C-P can effectively reduce the total reward of a training agent, or force the agent to choose a specific policy with limited power and budget. Moreover, VA2C-P works for heterogeneous poison aims, and works in both white-box and black-box settings.

**Experiment Setup.** We choose 4 policy-gradient learning algorithms, including Vanilla Policy Gradient (Sutton et al., 2000), A2C (Mnih et al., 2016), ACKTR (Wu et al., 2017) and PPO (Schulman et al., 2017). And we choose 5 Gym (Brockman et al., 2016) environments with increasing difficulty levels: CartPole, LunarLander, Hopper, Walker and HalfCheetah. All results are averaged over 10 random seeds. The total effort $U$ is calculated by the normalized $\ell_2$-distance. See Appendix G.1 for the expression of $U$, as well as more hyper-parameter settings.

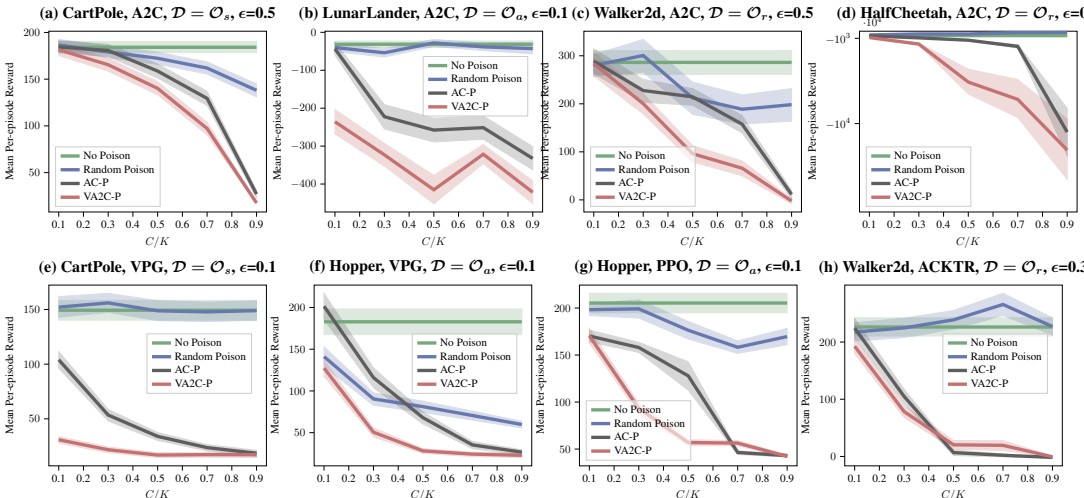

**Figure 3:** Comparison of mean per-episode reward gained by VPG, PPO, A2C, ACKTR on various environments, under no poisoning, random poisoning, AC-P and VA2C-P.

**Reward-minimizing Poisoning.** *Baselines.* To the best of our knowledge, there is *no existing poisoning algorithm against deep policy-gradient algorithms*. Although some evasion methods (Inkawhich et al., 2020) also work for policy-gradient algorithms, evasion is substantially different from poisoning since it does not influence the policy. Therefore, to show the effectiveness of VA2C-P, we compare it with 3 baselines: (1) the normal learning with no poison; (2) a random attacker which randomly chooses $C$ iterations, and perturbs the reward to an arbitrary direction by $\epsilon$; and (3) a simplified version of our algorithm, called Adversarial Critic Poison (AC-P), which decides "how to attack" in the same way as VA2C-P does, but chooses "when to attack" randomly.

*Performance.* We first show the reward-minimizing performance of VA2C-P on all three types of poison aims, assuming the attacker knows the learner's model (white-box attack). Figure 3 shows the rewards of various learners under different kinds of poisoning methods, with different ratios of budget $C$ to the total number of iterations $K$. Compared with random poisoning, our proposed *VA2C-P and the simplified version AC-P make the reward drop more significantly*, demonstrating the effectiveness of our "how to attack" decisions made by the Adversarial Critic. VA2C-P further outperforms AC-P in almost all cases, which implies that our "when to attack" decisions based on Vulnerability-Awareness work well in practice. An interesting observation is *random poisoning not only does not work well in many cases, but sometimes also facilitates the learner* (Figure 3h). This

phenomenon is mainly due to the uncertainty of the environment, as pointed out by *Challenge I* and *Challenge II* in Section 1. Thus, a good poisoning strategy is important.

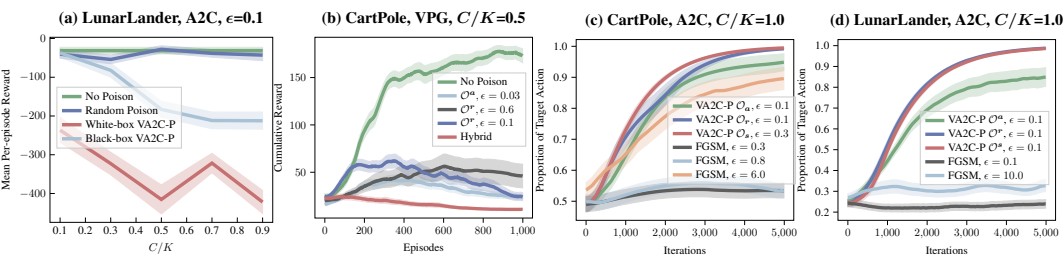

**Figure 4:** (a) VA2C-P also works under black-box setting; (b) hybrid-aim poisoning could be better than single-aim poisoning; (c)(d) VA2C-P successfully forces the agent to choose the target policy.

**Black-box Poisoning.** Figure 3 demonstrates the performance of VA2C-P when the attacker knows the learner's model. But when the learner's model is not available (black-box attack), VA2C-P could also work as shown in Figure 4a, where one can see that *black-box poisoning is still effective, although worse than white-box poisoning due to the lack of knowledge.*

**Hybrid-aim Poisoning.** The experiments shown in Figure 3 assume that the attacker poisons one of the 3 poison aims, $\mathcal{O}^s, \mathcal{O}^a$ or $\mathcal{O}^r$. But if the attacker happens to have access to all of these poison aims, it can do better by performing a "hybrid" poisoning, i.e., at each iteration, evaluate the vulnerability of the algorithm w.r.t. each poison aim with their corresponding attacker power, then choose the one with the highest vulnerability. Figure 4b shows that *adaptive attacking using a hybrid poison aim could significantly outperform attacking using a fixed single poison aim.*

**Targeted Poisoning.** *Baselines.* Although targeted RL poisoning is studied by many works (Huang & Zhu, 2019; Rakhsha et al., 2020), few of them work for deep RL. Despite that no existing work focuses on deep policy-based methods, we transfer the FGSM-based targeted poisoning method proposed for DQN by Behzadan & Munir (2017) to attack policy-based learners as our baseline. The attacking method is, for each new state, adding a perturbation to it such that the perturbed state is pushed across the decision boundary and towards the target action. See Appendix G.2 for details.

*Performance.* Figure 4c and 4d respectively show the targeted-attack performance comparison on CartPole and LunarLander, where the target policies are both "always going to the right". The y-axes show the proportion of target actions among all actions taken by the learner. *VA2C-P successfully leads the learner to learn the target policy using all types of poison aims with relatively small attack power.* In contrast, FGSM fails to let the learner select the target action in most cases, even with much larger attack power. The main reason why FGSM works for DQN Behzadan & Munir (2017) but not for policy-gradient methods is, policy-based agents generally learn stochastic policies, and even though FGSM could perturb the state such that the output probability for the target action is 0.51, the agent will still choose other actions with probability 0.49. Therefore, *FGSM cannot easily perform targeted poisoning against policy-based learners.*

More experiment results are in Appendix G.3. Note that our proposed poisoning method can also be extended to off-policy learners, as discussed in Appendix G.4.

## 6 CONCLUSION AND DISCUSSION

In this paper, we propose VA2C-P, the first generic poisoning algorithm for deep policy-gradient online RL methods, which incorporates heterogeneous poisoning models and does not require any prior knowledge of the environment. Although the effectiveness of VA2C-P is verified in a wide range of RL algorithms and environments, we acknowledge that poisoning RL in practice is still challenging, due to the relatively high computational burden and the uncertainty of the environments. These challenges are also exciting opportunities for future work.

### ACKNOWLEDGMENTS

This work is supported by National Science Foundation IIS-1850220 CRII Award 030742-00001 and DOD-DARPA-Defense Advanced Research Projects Agency Guaranteeing AI Robustness against Deception (GARD), and Adobe, Capital One and JP Morgan faculty fellowships.

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

# Appendix: Vulnerability-Aware Poisoning Mechanism for Online RL with Unknown Dynamics

## A ADDITIONAL RELATED WORKS

**Adversarial Attacks in SL.** There have been many works studying adversarial attacking and defending in the past decade (Huang et al., 2011; Steinhardt et al., 2017). Poisoning, as an important type of adversarial attacking, has been well-studied in the field of machine learning (Biggio et al., 2012; Mei & Zhu, 2015), including deep learning (Muñoz-González et al., 2017; Shafahi et al., 2018). Wang & Chaudhuri (2018) consider online learning, which is similar to ours. However, they focus on supervised learning, where the attacker knows all the data stream, while in our RL setting, the whole training data stream is not available to the attacker (as Challenge I states).

**Evasion Attacks in RL.** Recently, adversarial RL is attracting more and more attention, especially evasion attacks at test-time, as summarized by Chen et al. (2019). Most works consider adversarial perturbations on observations, similar to adversarial examples in supervised learning. Huang et al. (2017) first show that neural network policies are vulnerable to evasion attacks on states, by generating state perturbation with FGSM. Lin et al. (2017) consider the data-correlation problem in RL and an attacker with limited ability, and propose a strategical attack method, which perturbs input states only under certain conditions. In addition, Gleave et al. (2019) show that in multi-agent RL, choosing an adversarial policy could also negatively affect the victim agent, which is similar to perturbing a state into a novel one. There are plenty of following works showing the vulnerability of deep RL policies, even against a limited-power black-box attacker Xinghua et al. (2020); Inkawhich et al. (2020), Adversarial attacks can be utilized to train robust policies Pinto et al. (2017); Pattanaik et al. (2018). There is also a line of work focusing on adversarial examples in path-finding problems Xiang et al. (2018); Bai et al. (2018). These evasion attacks are created to fool a well-trained policy, but they can not change the policy itself in training time as poisoning does.

## B A POISONING FRAMEWORK FOR RL

In this section, we establish a generic framework of poisoning in online RL, systematically characterizing its challenges and difficulties from multiple perspectives - objective of poisoning, various poison aims, and attacker's knowledge. Our in-depth comparison with the SL allows a thorough understanding of the additional vulnerability of online RL systems compared with the well-understood SL systems. Our framework also provides a clear context to correctly position prior works in the literature as well as to compare our work with existing works. Compared to the poisoning framework described by Huang & Zhu (2019), we provide a solution for unifying these poison aims in one attack model in this section.

We consider the online learning scenario, where the RL agent (the learner) does not know the dynamics or rewards of the underlying MDP $\mathcal{M}$ with state space $\mathcal{S}$, action space $\mathcal{A}$, transition dynamics $P$, rewards $R$ and discount factor $\gamma$.

**Settings and Notations** In online RL, the learner interacts with the environment and collects observations. The learner's algorithm, denoted by $f$, iteratively searches for a policy $\pi$ parametrized by $\theta$, through $K$ interactions with the environment. Before learning starts, the learner initializes a policy $\pi_1$. At each iteration $k$, the learner uses its previous policy $\pi_{k-1}$ to roll out *observations* $\mathcal{O}_k$ from the MDP $\mathcal{M}$. $\mathcal{O}_k$ is a concatenation of multiple *trajectories*, denoted as $\mathcal{O}_k = (\boldsymbol{s}_k, \boldsymbol{a}_k, \boldsymbol{r}_k)$, where $\boldsymbol{s}_k = [s_1, s_2, \cdots], \boldsymbol{a}_k = [a_1, a_2, \cdots], \boldsymbol{r}_k = [r_1, r_2, \cdots]]$ are respectively the sequence of states, actions, and rewards in iteration $k$. Then, with the observations $\mathcal{O}_k$, the learner updates its policy by attempting to solve $\operatorname{argmax}_\pi J(\pi, \pi_{k-1}, \mathcal{O}_k)$, where $J$ is the objective function. The generated policy by the learner's algorithm $\pi_k = f(\pi_{k-1}, \mathcal{O}_k)$[4] does not necessarily achieve the maximization of the objective function.

In this paper, an overhead check sign ˘ on a variable always denotes that the variable is poisoned. For example, if the attacker changes a reward $r_t$, then the poisoned reward is denoted as $\breve{r}_t$.

---

[4]For algorithms with experience replay, the update can be extended as $\pi_{k+1} = f(\pi_k, \mathcal{O}_{1:k})$.

---

**Procedure 2:** Flow of Online RL Poisoning

---

1 Learner initializes its initial policy $\pi_{\theta_0}$
2 **for** $k = 1, \cdots, K$ **do**
3     Learner rollouts observation (trajectories) $\mathcal{O}_k$ in environment $\mathcal{M}$ with current policy $\pi_k$
4     Attacker may poison the observation $\mathcal{O}_k$ to $\check{\mathcal{O}}_k$
5     Learner updates its policy: $\pi_{k+1} = f(\pi_k, \check{\mathcal{O}}_k) \approx \text{argmax}_\pi J(\pi, \pi_k, \check{\mathcal{O}}_k)$

---

**Poison Objective.** We use $L_A$ to denote loss function of the poisoning attack, which the attacker attempts to minimize. The form of $L_A$ is determined by its goal, which falls into one of the two categories, *non-targeted* and *targeted* poisoning.

In *non-targeted poisoning*, the attack poisons a policy $\pi$ to $\check{\pi}$ to minimize the learner's expected rewards. Therefore the poison objective $L_A$ is to minimize the learner's value $\eta(\check{\pi})$.

In *targeted poisoning*, the attack "teaches" the agent to learn a pre-defined target policy $\pi^\dagger$. Therefore the poison objective $L_A$ is defined as distance[5] $(\check{\pi}, \pi^\dagger)$.

Most existing poison RL researches focus on targeted poisoning (Ma et al., 2019; Rakhsha et al., 2020), and non-targeted poisoning, although discussed by Huang & Zhu (2019), remains relatively untouched.

**poison aims.** To influence the behaviors of the learner, an attacker could inject poison at multiple locations of the learner's learning process as detailed in Figure 5. Part of the reason why poisoning in RL is more challenging than in SL is that it involves more poison aims of poisoning, some of which adapt with the environment, increasing the uncertainty.

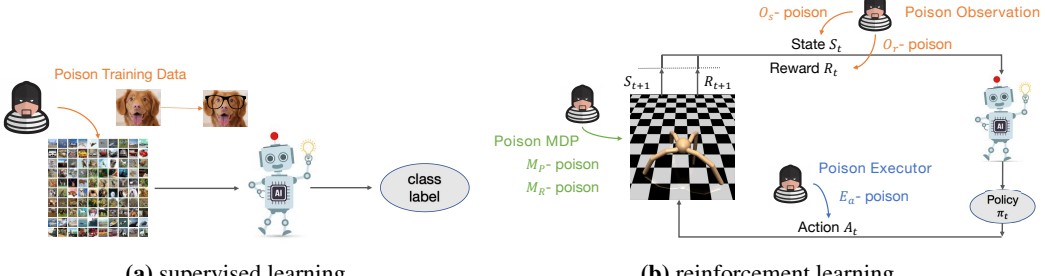

| **(a)** supervised learning | **(b)** reinforcement learning |

**Figure 5:** Different poison aims of poisoning in supervised learning and reinforcement learning.

*Poison Aim I – Poison Observation $(\mathcal{O}_r, \mathcal{O}_s)$.* The attacker could manipulate the observation of the learner, i.e., change $\mathcal{O}$ into $\check{\mathcal{O}}$. This may happen when the attacker is able to intercept the communication between the learner and the environment, similar to the man-in-the-middle attack in cryptography. The attacker could target the rewards, called $\mathcal{O}_r$-*poisoning*, studied by Huang & Zhu (2019); or the states, called $\mathcal{O}_s$-*poisoning*, investigated by Behzadan & Munir (2017).

*Poison Aim II – Poison MDP $(\mathcal{M}_R, \mathcal{M}_P)$.* An attack could directly change the MDP (environment) that the learner is interacting with, i.e., change $\mathcal{M}$ into $\check{\mathcal{M}}$. For example, a seller could influence the behaviors of customers by changing the prices of products. The poison of MDP could be injected at the reward model $R$ or the transition dynamics $P$, respectively denoted as $\mathcal{M}_R$-*poisoning* (studied by Ma et al. (2019)) and $\mathcal{M}_P$-*poisoning* (studied by Rakhsha et al. (2020)). The analogy of poison MDP in SL is to manipulate the underlying data distribution of the training data.

*Poison Aim III – Poison Executor $E_a$.* The executor of the learner could be poisoned. For example, an attacker applies a force to the agent, so that the intended action "north" becomes "northeast". Pinto et al. (2017) train a robust RL agent against the executor poisoner. Denote this type of poisoning as $E_a$-*poisoning*. We show in the Appendix C that $E_a$-poisoning is equivalent to directly changing $a$ stored in the observation $\mathcal{O} = (s, a, r, d)$.

---

[5]There are many ways to define the distance between two policies, for instance KL-divergence for stochastic policies (Schulman et al., 2015a), and average mismatch for deterministic policies (Rakhsha et al., 2020).

**Attacker's Knowledge** At the $k$-th iteration, what an attacker can do depends on its current knowledge set, denoted by $\mathcal{K}_k$. $\mathcal{K}_k$ could contain the underlying MDP $\mathcal{M}$, the learner's algorithm $f$, the learner's previous policy models $\theta_{1:k-1}$ as well as the previous and current observations $\mathcal{O}_{1:k}$.

An *omniscient attacker* knows everything , i.e, $\mathcal{K}_k^{(O)} = \{\mathcal{M}, f, \theta_{1:k-1}, \mathcal{O}_{1:k}\}$. Most guaranteed policy teaching literature (Rakhsha et al., 2020; Ma et al., 2019) assume omniscient attacker. However as motivated in the introduction, it is often unrealistic to exactly know the underlying environment. We discuss two more realistic setting where the attacker only has limited knowledge as follows.

A *monitoring attacker* has some information but does not assume knowledge of the underlying MDP $\mathcal{M}$, i.e., $\mathcal{K}_k^{(M)} = \{f, \theta_{1:k-1}, \mathcal{O}_{1:k}\}$. This is especially relevant in applications where learner's information is not secure (or even open), or an attacker hacks to steal information from the learner. Monitoring attacker is similar to the white-box attacker in supervised learning.

A *tapping attacker* has very limited knowledge and knows the observations only, i.e., $\mathcal{K}_k^{(T)} = \{\mathcal{O}_{1:k}\}$. This is widely applicable since the tapping the communication between the learner and the environment is easy. Tapping attacker is analogous to the black-box attacker in supervised learning. Behzdan & Munir (2017) consider a tapping attacker, which observes and manipulates the states but does not know the learner's parameters.

## C    TARGET TYPES OF ATTACKING: MORE DETAILED EXPLANATION

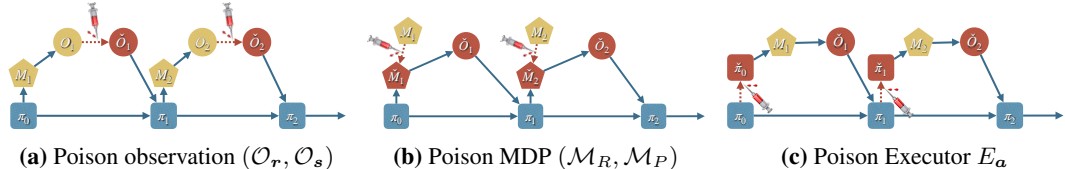

**(a)** Poison observation $(\mathcal{O}_r, \mathcal{O}_s)$         **(b)** Poison MDP $(\mathcal{M}_R, \mathcal{M}_P)$         **(c)** Poison Executor $E_a$

**Figure 6:** The online poisoning vs learning. Blue solid lines denote the learning processes, while red dashed lines denote the poisoning processes. In iteration $k$, the learner uses its previous policy $\pi_{k-1}$ to roll out observations $\mathcal{O}_k$ from current MDP $\mathcal{M}_k$, then updates its model and policy by $\pi_k = f(\pi_{k-1}, \mathcal{O}_k) = \arg\max_\pi J(\pi, \pi_{k-1}, \mathcal{O})$. The attacker may (a) poison observations after they are generated, (b) poison MDP before the learner generates observations, or (c) poison the policy when it is used to generate observations. Among these three scenarios, $\mathcal{O}_k$ is always influenced by the attack, thus denoted as $\check{\mathcal{O}}_k$.

### C.1    POISON OBSERVATION

Consider a deep reinforcement learning algorithm, which uses its old policy to generate a series of observations and updates its policy with the collected observations at every iteration. The observations are stored in a temporary buffer in the form of $(s, a, r, d)$, where $s, r, d$ are returned by the environment, and $a$ is produced by the policy itself.

An attacker could stay in the middle between the learner and the environment and falsify $s, r$ returned by the environment before the learner receives them. (It is also possible to alter the terminal state flags $d$, but it is relatively easier for the learner to detect, and its influence to the learner is not as large as $s$ and $r$, so we do not discuss this attack type.) On the other hand, the attacker may also hack the observation buffer to change $s, r$. In both cases, poisoning $s$ and $r$ are called $\mathcal{O}_s$-poisoning and $\mathcal{O}_r$-poisoning respectively.

Note that in the case of hacking observation buffer, the attacker also has the option to manipulate $a$ in $\mathcal{O}$. But we do not include this kind of attack in observation poisoning, because not like states and rewards, the actions are taken by the learner, so it is easy for the learner to detect the change of action sequence stored in the buffer. And we will show in Section C.3 that changing $a$ in $\mathcal{O}$ is similar to the case of executor poisoning.

## C.2 Poison MDP

MDP poisoning can be considered as "changing the reality" for the learner. For example, an attacker decides to increase the reward for state-action pair $(s, a)$ by $\Delta$ at iteration $t$, it then changes the parameter $R_k(s, a)$ of the environment to $\check{R}_k(s, a) = R_k(s, a) + \Delta$. As a result, whenever the learner visits $(s, a)$ in the current iteration, it receives $\check{R}_k(s, a)$ as its reward. A more intuitive example is depicted in Figure 7, where we want to train a mouse to find the cheese in a maze. However, if some bad man adds another piece of cheese in the training environment, the mouse is likely to be misled. Then, in the test time, the "malicious cheese" is removed, but the mouse still wants to find the malicious cheese instead of the original one.

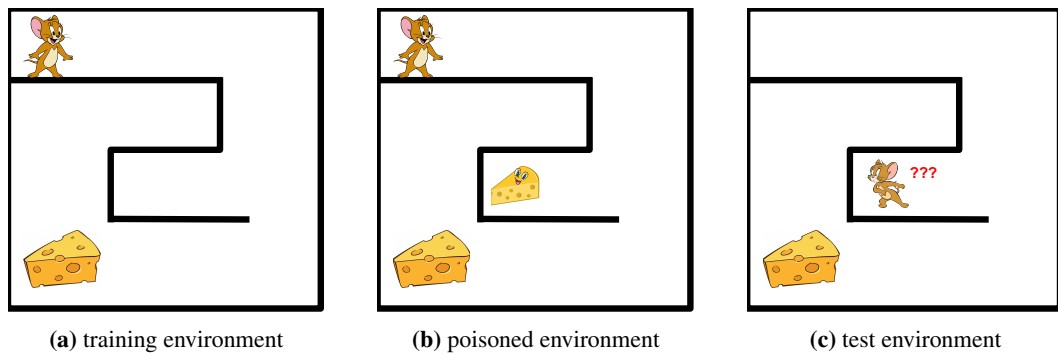

    **(a)** training environment         **(b)** poisoned environment         **(c)** test environment

**Figure 7:** An intuitive example of MDP poisoning.

Compared with observation poisoning and executor poisoning, MDP Poisoning is more powerful and more difficult to defend against. An example of MDP poisoning is given by Rakhsha et al. (2020), where the attacker can force the learner to learn a target policy with guarantees.

## C.3 Poison Executor

In the online RL process, the learner learns by taking actions and getting feedbacks of the actions. If observation poisoning is viewed as changing the feedbacks, then executor poisoning can be viewed as perturbing the actions taken by the learner. For example, for an auto-driving agent, one can slightly add some force to the steering wheel, so when the agent takes "steer left 90 degrees", what actually happens is "steer left 100 degrees". In this way, the trained policy is biased.

We now show that the aforementioned attack of changing $\boldsymbol{a}$ in $\mathcal{O}$ could be converted to executor poisoning. We describe the following two scenarios respectively for these two poisoning attacks and show they have equal effects. For simplicity, we only consider one step of the interaction, which can be simply extended to multiple steps.

**(1) Poisoning $\boldsymbol{a}$ in $\mathcal{O}$.** For one-step experience, the learner observes state $s$, takes action $a_i$, receives reward $r_i = R(s, a_i)$ and observes a new state $s_i' \sim P(\cdot|s, a_i)$, then the tuple $(s, a_i, r_i, s_i')$ will be stored in the observation buffer. Now the attacker may change the action $a_i$ to $a_j$, and the tuple becomes $(s, a_j, r_i, s_i')$. Finally, the learner regard $r_i, s_i'$ and the following observations as caused by $a_j$, instead of $a_i$.

**(2) Poisoning Executor.** In the same environment, suppose the learner observes state $s$, and takes action $a_j$, then the attacker conducts executor poisoning and changes $a_j$ to $a_i$, then the reward and the next state will also change to $r_i = R(s, a_i)$ and $s_i' \sim P(\cdot|s, a_i)$. The learner does not know the falsification of action and stores $(s, a_j, r_i, s_i')$ to the buffer. Finally, the attacker will take $r_i, s_i'$ and the future observations as caused by $a_j$, and updates its policy accordingly.

Based on the descriptions, we can see changing $a_i$ to $a_j$ in the buffer is equivalent to changing $a_j$ to $a_i$ on the executor. If we assume the policy always has non-zero probability on all actions, then for any manipulation on $\boldsymbol{a}$ in $\mathcal{O}$, it is possible to achieve the same effects by attacking the executor. Hence, given the fact that the former attack can be trivially defended, we only discuss executor poisoning in our poisoning framework.

# D  ROBUSTNESS AND STABILITY OF RL ALGORITHMS

## D.1  MEASURING VULNERABILITY OF RL ALGORITHMS

Definition 1 in Section 4 measures the stability of one update based on policy discrepancy. But it is not obvious whether the rewards change drastically due to the attacks. Proposition 2 provides a guarantee on the performance of the poisoned policy $\tilde{\pi}'$, compared with the un-poisoned new policy $\pi'$.

**Proposition 2.** *For an update of stochastic policy $\pi' = f(\pi, \mathcal{O})$, if its $\delta$-stability radius is $\varepsilon$ with the total variance measure, then any poisoning effort smaller than $\varepsilon$ on $\mathcal{D}$ will cause the expected total reward drop $(\eta(\pi') - \eta(\tilde{\pi}'))$ by no more than*

$$\frac{4\delta^2 \gamma \max_{s,a} |A_{\pi'}(s,a)|}{(1-\gamma)^2} + 2\delta \sum_{s \in \mathcal{S}} g_{\pi'}(s) \max_a A_{\pi'}|(s,a)| \tag{2}$$

*where $\gamma$ is the discount factor, $g_\pi(s) := \sum_{t=0}^{\infty} \gamma^t P(s_t = s|\pi)$ is the discounted visitation frequency, and $A$ is the advantage function, i.e., $A_\pi(s,a) = Q_\pi(s,a) - V_\pi(s)$.*

Proposition 2 shows that if the attack power is within the stability radius, the reward of the poisoned policy will not be influenced too much. which also explains the motivation behind our proposed vulnerability-aware attack. The proof is in Appendix D.4.

With Definition 1, the one-update stability measure, we are able to formally define the stability radius of an RL algorithm w.r.t. an MDP.

**Definition 3** (Stability Radius w.r.t an MDP). *The $\delta$-stability radius of an algorithm $f$ in an MDP $\mathcal{M}$ is defined as the minimal stability radius of all observations drawn from the MDP, and all possible policies in policy space $\Pi$. If $f$ is on-policy, then $\phi(f, \mathcal{M}) = \min_{\pi \in \Pi, \mathcal{O} \sim \pi} \phi(f, \pi, \mathcal{O})$; if $f$ is off-policy, then $\phi(f, \mathcal{M}) = \min_{\pi \in \Pi, \mathcal{O} \sim \overline{\pi}} \phi(f, \pi, \mathcal{O})$, where $\overline{\pi}$ is the behavior policy;*

## D.2  ROBUSTNESS RADIUS

Different with poisoning, test-time evasion (adversarial examples) misleads the agent by manipulating the states only, since the agent no longer learns from interactions and feedbacks. Note that although Gleave et al. (2019) propose an attack called "adversarial policy", the perturbation does not happen in the policy, but still happens in the input states (observations) of the agent.

To study how robust a trained policy is, we define the robustness radius with regard to both a single state and the whole environment.

**Definition 4** (Robustness Radius of Policy w.r.t. a State). *The robustness radius of a deterministic policy $\pi$ on a state $s$ is defined as the minimal perturbation of $s$ which changes the output action, i.e.,*

$$\rho(\pi, s) = \inf_\varepsilon \{\exists \check{s} \in \mathcal{S} \cap \mathcal{B}_\varepsilon(s) \quad s.t. \quad \pi(s) \neq \pi(\check{s})\} \tag{3}$$

*Similarly, for any $0 < \delta < 1$, the $\delta$-robustness radius of a stochastic policy $\pi$ on a state $s$ is defined as the minimal perturbation of $s$ which makes the output action distribution disagrees with the original $\pi(s)$ with probability more than $\delta$, i.e.,*

$$\rho(\pi, s)_\delta = \inf_\varepsilon \{\exists \check{s} \in \mathcal{S} \cap \mathcal{B}_\varepsilon(s) \ s.t. \ d(\pi(\cdot|s)||\pi(\cdot|\check{s})) > \delta\} \tag{4}$$

*where $d(\cdot|\cdot)$ could be any distance measure between two distributions.*

**Remark.** If we regard the policy as a classifier which "classifies" a state to an action, then the robustness radius of policy defined above is analogous to the robustness radius of classifiers defined by Wang et al. (2017), with an extension to stochastic predictions.

**Definition 5** (Robustness Radius w.r.t an MDP). *The $(\delta$-$)$robustness radius of a policy $\pi$ in an MDP $\mathcal{M}$ is defined as the maximal robustness radius of all states, i.e., $\rho(\pi, \mathcal{M}) = \min_{s \in \mathcal{S}} \rho(\pi, s)$*

**Remarks.** (1) A deterministic policy is robust against any state perturbation smaller than $\rho(\pi, \mathcal{M})$.

(2) For a stochastic policy, if its $\delta$-robustness radius in an $\mathcal{M}$ is $\varepsilon$, then any state perturbation within $\varepsilon$ will cause the expected total reward drop by no more than

$$(\frac{2\delta\gamma}{(1-\gamma)^2} + 2\delta) \max_{s,a} |R(s,a)|, \tag{5}$$

which is proven by Theorem 5 in Zhang et al. (2020a).

### D.3 VULNERABILITY COMPARISON: DIFFERENCE BETWEEN SL AND RL

To shed some light on understanding adversarial attacks in RL, we compare SL and RL in terms of their vulnerability to poisoning and adversarial examples.

At test time, a policy network receives states as input, and returns probabilities of choosing each actions as output; a value network receives states (or state-action pairs) as input, and returns the corresponding value as the output. Thus, test-time RL systems are very similar to SL systems, as one can view the policy networks as classification networks, and value networks as regression networks. However, the key difference between evasion in RL and evasion in SL is, data samples are not independent in RL. A single adversarial example in SL test dataset may cause at most one misclassification instance, whereas an adversarial example in RL may case a drastic change of the gained rewards (e.g., by leading the agent to a "devastating" or "absorbing" state).

At training time, SL systems and RL systems are significantly different, as Figure 5 shows. Even when the supervised learner also learns from data streams in an online manner, the training data are independent with the learner's classifier. In contrast, the distribution of training data samples changes as the learner updates its policy. Poisoning attacks against an SL system could alter the decision boundary, so that the learner makes wrong decisions for certain data samples. For an RL system, poisoning attacks could (1) alter the decision boundary so that the learner chooses bad actions for certain states, and also (2) change the following observations and interactions due to a different selection of action.

In summary, an adversarial attacker may cause higher damages on RL systems than on SL systems, with the same power and budget. But it does not suggests attacking RL systems is easier than attacking SL systems. As every coin has two sides, the high uncertainty of the environment may help an attack reduce the learner's reward, but may also lead the learner to gain higher reward in the future (as shown in Section 5). Therefore, it is more challenging to successfully attack RL systems than SL systems with a specific goal.

### D.4 PROOF OF PROPOSITION 2

*Proof.* According to the definition of $\delta$-stability radius, for any poisoning effort within $\varepsilon$, the poisoned policy satisfies $D_{TV}^{\max}[\pi'||\check{\pi}'] \leq \delta$ (assume total variance $D_{TV}$ is the distance measure between policy distributions). We are interested in the difference of expected rewards $\eta(\pi') - \eta(\check{\pi}')$.

Define $L_{\pi'}(\check{\pi}') = \eta(\pi') + \sum_{s \in \mathcal{S}} g_{\pi'}(s) \sum_{a \in \mathcal{A}} \check{\pi}'(a|s) A_{\pi'}(s,a)$, where $g$ is the discounted state visitation frequencies, i.e.,

$$g_{\pi'}(s) := P(s_0 = s|\pi') + \gamma P(s_1 = s|\pi') + \gamma^2 P(s_2 = s|\pi') + \cdots.$$

Since $D_{TV}^{\max}[\pi'||\check{\pi}'] \leq \delta$, follow Theorem 1 in paper (Schulman et al., 2015a), we can get

$$|\eta(\check{\pi}') - L_{\pi'}(\check{\pi}')| \leq \frac{4\delta^2\gamma \max_{s,a} |A_{\pi'}(s,a)|}{(1-\gamma)^2}. \tag{6}$$

So we have

$$|\eta(\check{\pi}') - \eta(\pi') - \sum_{s \in \mathcal{S}} g_{\pi'}(s) \sum_{a \in \mathcal{A}} \check{\pi}'(a|s) A_{\pi'}(s,a)| \leq \frac{4\delta^2\gamma \max_{s,a} |A_{\pi'}(s,a)|}{(1-\gamma)^2}, \tag{7}$$

which can be transformed to

$$\eta(\pi') - \eta(\check{\pi}') \leq \frac{4\delta^2\gamma \max_{s,a} |A_{\pi'}(s,a)|}{(1-\gamma)^2} - \sum_{s \in \mathcal{S}} g_{\pi'}(s) \sum_{a \in \mathcal{A}} \check{\pi}'(a|s) A_{\pi'}(s,a). \tag{8}$$

We upper bound the term $-\sum_{s\in\mathcal{S}} g_{\pi'}(s) \sum_{a\in\mathcal{A}} \check{\pi}'(a|s) A_{\pi'}(s,a)$ as below.

$$
\begin{aligned}
&-\sum_{s\in\mathcal{S}} g_{\pi'}(s) \sum_{a\in\mathcal{A}} \check{\pi}'(a|s) A_{\pi'}(s,a) \\
=&\sum_{s\in\mathcal{S}} g_{\pi'}(s)\Big(-\sum_{a\in\mathcal{A}} \check{\pi}'(a|s) A_{\pi'}(s,a)\Big) \\
=&\sum_{s\in\mathcal{S}} g_{\pi'}(s)\Big(-\sum_{a\in\mathcal{A}} \check{\pi}'(a|s) A_{\pi'}(s,a) + \sum_{a\in\mathcal{A}} \pi'(a|s) A_{\pi'}(s,a) - \sum_{a\in\mathcal{A}} \pi'(a|s) A_{\pi'}(s,a)\Big) \\
=&\sum_{s\in\mathcal{S}} g_{\pi'}(s)\Big(\sum_{a\in\mathcal{A}} A_{\pi'}(s,a)(\pi'(a|s) - \check{\pi}'(a|s)) - \sum_{a\in\mathcal{A}} \pi'(a|s) A_{\pi'}(s,a)\Big) \\
\leq&\sum_{s\in\mathcal{S}} g_{\pi'}(s)\Big(2\delta \max_{a\in\mathcal{A}} |A_{\pi'}(s,a)| - \sum_{a\in\mathcal{A}} \pi'(a|s) A_{\pi'}(s,a)\Big) \\
=&2\delta \sum_{s\in\mathcal{S}} g_{\pi'}(s) \max_{a\in\mathcal{A}} |A_{\pi'}(s,a)|
\end{aligned}
\tag{9}
$$

Combining the above results, we obtain

$$
\eta(\pi') - \eta(\check{\pi}') \leq \frac{4\delta^2\gamma \max_{s,a} |A_{\pi'}(s,a)|}{(1-\gamma)^2} + 2\delta \sum_{s\in\mathcal{S}} g_{\pi'}(s) \max_a |A_{\pi'}(s,a)|
\tag{10}
$$

$\square$

# E  ALGORITHM

Assume the learner's policy $\pi$ is parametrized by $\theta$.

**How to solve the projected gradient descent.** To solve (P), assume $\frac{\partial\theta_k}{\partial\check{r}}$ exists, one can use projected gradient descent to update $r$ by using the chain rule:

$$
\frac{\partial\eta_{\pi_{\theta_{k-1}}}(\pi_{\theta_k})}{\partial\check{r}} = \frac{\partial\eta_{\pi_{\theta_{k-1}}}(\pi_{\theta_k})}{\partial\theta_k} \frac{\partial\theta_k}{\partial\check{r}}.
$$

For Vanilla Policy Gradient (VPG) whose update rule is, $\theta_k = \theta_{k-1} + \alpha\nabla_{\theta_{k-1}}\hat{\eta}(\pi_{\theta_{k-1}}, r)$, where

$$
\nabla_{\theta_{k-1}}\hat{\eta}(\pi_{\theta_{k-1}}, r) = \frac{1}{N}\sum_{i=1}^{N}\Big(\big(\sum_{t=1}^{T}\nabla_{\theta_{k-1}}\log\pi_{\theta_{k-1}}(a_t^{(i)}|s_t^{(i)})\big)\big(\sum_{t=1}^{T} r_t^{(i)}\big)\Big),
\tag{11}
$$

we can derive

$$
\nabla_\theta\eta_{\pi_{\theta_{k-1}}}(\pi_{\theta_k}) \approx \frac{1}{N}\sum_{i=1}^{N}\Big(\Pi_{t=1}^{T}\frac{\pi_{\theta_k}(a_t^{(i)}|s_t^{(i)})}{\pi_{\theta_{k-1}}(a_t^{(i)}|s_t^{(i)})}\big(\sum_{t=1}^{T}\nabla_{\theta_k}\log\pi_{\theta_k}(a_t^{(i)}|s_t^{(i)})\big)\big(\sum_{t=1}^{T}\gamma^{t'-1} r_t^{(i)}\big)\Big).
\tag{12}
$$

and

$$
[\nabla_r\theta_k]_t = \sum_{j=1}^{t}\nabla_{\theta_{k-1}}\log\pi_{\theta_{k-1}}(a_j|s_j)\gamma^{t-j}
\tag{13}
$$

Although $\frac{\partial\theta_k}{\partial\check{r}}$ has a closed-form expression for simple learners like VPG, analytically computing how the poisoned reward influences the model is challenging for more complicated learners like PPO, whose update rule is an argmax function. Therefore, we use the Direct Gradient Method proposed by Yang et al. (2017) to approximate the gradient by

$$
\frac{\partial\eta_{\pi_{\theta_{k-1}}}(\pi_{\theta_k})}{\partial\check{r}} = \frac{\eta_{\pi_{\theta_{k-1}}}(f(\pi_{\theta_{k-1}}, r+\Delta)) - \eta_{\pi_{\theta_{k-1}}}(f(\pi_{\theta_{k-1}}, r))}{\Delta}
\tag{14}
$$

---

**Algorithm 3:** Non-targeted White-box VA2C-P with $\mathcal{O}_{\boldsymbol{r}}$-Poisoning

---

**Input:** total iterations of learning $K$; poisoning power $\epsilon$; poisoning budget $C$; attacker's
   learning rate $\beta$; maximum computing iterations $J$; distribution distance measure $d$

1 Initialize policy discrepancies as an empty list $\Psi = \emptyset$
2 Initialize the number of already poisoned iterations $c = 0$
3 Initialize value network $V_\omega$
4 **for** $k = 1, \cdots, K$ **do**
5    **if** $c > C$ **then**
6       | Break
7    Get the current observation $\mathcal{O}_k$ and the learner's policy model $\theta_{k-1}$
8    Fit the value function: $\omega \leftarrow \operatorname{argmin}_\omega \sum_{\tau_i \in \mathcal{O}_k} \sum_{t=1}^T (V_\omega(s_t^{(i)}) - \sum_{t'=t}^T \gamma^{t'-t} r_t^{(i)})^2$
9    Imitate learner's update with the clean rewards $\theta_k, \eta \leftarrow \texttt{Update}(\theta_{k-1}, \boldsymbol{r})$
10    Initialize $\check{\boldsymbol{r}}$ as the original $\boldsymbol{r}$ in $\mathcal{O}_k$
11    Set $\eta_0 = \eta_c$
12    **for** $j = 1, \cdots, J$ **do**
13       **for** $i = 1, \cdots, N$ *and* $t = 1, \cdots, T$ **do**
14          Copy $\boldsymbol{r}' \leftarrow \check{\boldsymbol{r}}$, and add a small value $\Delta$ to $[r']_t^{(i)}$
15          Imitate learner's update with poisoned rewards $\theta', \eta' \leftarrow \texttt{Update}(\theta_{k-1}, \boldsymbol{r}')$
16          Compute the direct gradient: $\frac{\partial \eta}{\partial \check{\boldsymbol{r}}_t^{(i)}} \leftarrow \frac{\eta' - \eta_{j-1}}{\Delta}$
17       Update the poisoned reward: $\check{\boldsymbol{r}} \leftarrow \Pi_{\mathcal{B}_\epsilon(\boldsymbol{r})} (\check{\boldsymbol{r}} - \beta \frac{\partial \eta}{\partial \check{\boldsymbol{r}}})$
18       Imitate learner's update with poisoned rewards $\check{\theta}_k, \eta_j \leftarrow \texttt{Update}(\theta_{k-1}, \boldsymbol{r}')$
19       **if** $(\eta_j - \eta_{j-1})$ *converges* **then**
20          | Break
21    Compute $\psi_k = \frac{1}{NT} \sum_{s_t^{(i)}} d(\pi_{\theta_k} || \pi_{\check{\theta}_k})$ and add $\psi_k$ to $\Psi$
22    **if** $\psi_k$ *is larger than the* $\lfloor (C-c)/(K-k) \rfloor$*-th largest element in* $\Psi$ **then**
23       Attack: replace $\boldsymbol{r}$ with $\check{\boldsymbol{r}}$ in $\mathcal{O}_k$ and send it back to the learner
24       $c \leftarrow c + 1$
25 **Procedure** $\texttt{Update}(\theta, \boldsymbol{r})$
26    Perform an update with the learner's algorithm $\theta' \leftarrow f(\theta, \boldsymbol{r})$
27    Compute the attacker's objective

$$\eta \leftarrow \frac{1}{NT} \sum_{\tau_i \in \mathcal{O}_k} \sum_{t=1}^T \left( \frac{\pi_{\theta'}(a_t^{(i)}|s_t^{(i)})}{\pi_\theta(a_t^{(i)}|s_t^{(i)})} \right) \left( \sum_{t'=t}^T \gamma^{t'-t} r_t^{(i)} - V_\omega(s_t^i) \right)$$

28    **return** $\theta', \eta$

---

To show the concrete poisoning process, we assume $\mathcal{D} = \mathcal{O}^{\boldsymbol{r}}$. Then Algorithm 3 shows the detailed procedure of VA2C-P with a non-targeted goal and a white-box attacker.

**For Targeted Poisoning.** In line 16, instead of computing $\frac{\partial \eta}{\partial \check{\boldsymbol{r}}_t^{(i)}}$, we compute $\frac{\partial \operatorname{dist}(\theta', \theta^\dagger)}{\partial \check{\boldsymbol{r}}_t^{(i)}}$.

**For Black-box Poisoning.** In line 7, instead of getting the learner's policy model $\theta_{k-1}$, we train a policy with the same algorithm of the learner $\tilde{\theta}_{k-1} \leftarrow f(\tilde{\theta}_{k-2}, \mathcal{O}_{k-1})$.

## F    THEORETICAL INTERPRETATION OF THE BI-LEVEL OPTIMIZATION PROBLEM

In this section, we discuss the problem relaxation made in Section 4.2. For notation simplicity, we focus on the case $\mathcal{D} = \mathcal{O}^{\boldsymbol{r}}$.

### F.1    PROBLEM FORMS

Suppose the attacker has budget $C = K$. Then following the format of Problem equation Q, we could define the original $\mathcal{O}_{\boldsymbol{r}}$-poisoning optimization problem for the poisoning at the $k$-th iteration as Problem equation $P*$. Note that for notation simplicity, we use the policy parameter $\theta$ to denote

the policy $\pi_\theta$, so that $\eta(\theta) := \eta(\pi_\theta)$.

$$\underset{\check{r}_k, \cdots, \check{r}_K}{\operatorname{argmin}} \quad \sum_{j=k}^{K} \eta(\theta_j) \qquad (P*)$$
$$\text{s.t.} \qquad \theta_j = f(\theta_{j-1}, \check{r}_j)$$
$$\|\check{r}_j - r_j\| \leq \epsilon, \quad \forall j = k, \cdots, K$$

Note that we only optimize on $\check{r}_k$ to $\check{r}_K$, since previous $k - 1$ decisions have already been made at the $k$-th iteration.

Although an omniscient attacker is able to predict all the observations and solve Problem equation $P*$ directly, for the non-omniscient attacker that we focus on, Problem equation $P*$ is not solvable because of the unknown observations in iteration $k + 1$ to $K$. Hence, as discussed in Section 4, we relax equation $P*$, a multi-variable optimization problem, to $(K - k + 1)$ sequential single-variable optimization problems

$$\underset{\check{r}_k}{\operatorname{argmin}} \quad \eta(\theta_k) \qquad (P_k)$$
$$\text{s.t.} \qquad \theta_k = f(\theta_{k-1}, \check{r}_k)$$
$$\|\check{r}_k - r_k\| \leq \epsilon$$

$$\underset{\check{r}_{k+1}}{\operatorname{argmin}} \quad \eta(\theta_{k+1}) \qquad (P_{k+1})$$
$$\text{s.t.} \qquad \theta_{k+1} = f(\theta_k, \check{r}_{k+1})$$
$$\|\check{r}_{k+1} - r_{k+1}\| \leq \epsilon$$

and all the way to

$$\underset{\check{r}_K}{\operatorname{argmin}} \quad \eta(\theta_K) \qquad (P_K)$$
$$\text{s.t.} \qquad \theta_K = f(\theta_{K-1}, \check{r}_K)$$
$$\|\check{r}_K - r_K\| \leq \epsilon$$

where $r_j \sim \theta_{j-1}, \forall j = k, \cdots, K$.

Note that $r_k$ is already generated and known for all the above problems. When solving Problem $(P_j)$, $r_j$ and $\theta_j$ are also known (determined by $\theta_{j-1}$).

## F.2 Tightness of Problem Relaxation

Problem equation $P*$ is a $(K - k + 1)$-variable optimization problem with $(K - k + 1)$ equality constraints and $(K - k + 1)$ inequality constraints. And Problem equation $P_k, \cdots,$ equation $P_K$ are $(K - k + 1)$ single-variable optimization problems respectively with 1 equality constraints and 1 inequality constraints. The two sets of problems are naturally equivalent if $\theta_k, \cdots, \theta_K$, as well as the constraints are independent. However, due to the online learning process (Figure 6), $\theta_{k+1}$ and $r_{k+1}$ are all dependent on $\theta_k$ and $\check{r}_k$, which makes the relaxation not necessarily optimal.

We call the optimal solution to Problem equation $P*$ as $(\check{r}_k^*, \cdots, \check{r}_K^*)$, and the optimal solutions to Problem equation $P_k$ to equation $P_K$ as $\check{r}_k^\#, \cdots, \check{r}_K^\#$.

For simplicity, we assume the environment and the policies are all deterministic. So that the value of observed reward $r_j$ is a deterministic function of $\theta_{j-1}$.

We claim that the relaxation does not change the feasibility as stated in Proposition 6.

**Proposition 6.** $(\check{r}_k^\#, \cdots, \check{r}_K^\#)$ *is a feasible solution to Problem equation $P*$.*

*Proof.* Note that $r_k$ is known and the same for both equation $P*$ and equation $P_k$.

(1) $\check{r}_k^\#$ is feasible to equation $P*$ because it satisfies $\|\check{r}_k^\# - r_k\| \leq \epsilon$.

(2) For Problem equation $P*$, with $\theta_k = f(\theta_{k-1}, \check{r}_k^{\#})$, $r_{k+1}$ is the same with the $r_{k+1}$ in Problem $P_{k+1}$. Since the optimal solution to equation $P_{k+1}$, $\check{r}_{k+1}^{\#}$, satisfies $\|\check{r}_{k+1}^{\#} - r_{k+1}\| \leq \epsilon$, $\check{r}_{k+1}^{\#}$ is also feasible to equation $P*$.

(3) By induction, $\{\check{r}_j^{\#}\}_{j=k}^{K}$ all satisfy the constraints in equation $P*$ at the same time, so $(\check{r}_k^{\#}, \cdots, \check{r}_K^{\#})$ is a feasible solution to Problem equation $P*$. □

Note that if the environment or the policy is stochastic, then $r_j$ is a random variable sampled from some distribution defined by $\theta_{j-1}$. In this case, the constraints for Problem equation $P*$ should be $\Pr(\|\check{r}_j - r_j\| \leq \epsilon) \geq t, \forall j = k, \cdots, K$, where $t \in [0, 1]$ is a threshold probability. Then, with appropriate $t$, Proposition 6 also holds.

So far we have shown that the relaxation is feasible, so that the attacker will not plan for a non-feasible poisoning with VA2C-P. Next, we discuss the optimality of the relaxation.

We first make the following assumptions.

*Assumption 1*: the learner's update function $f(\theta, \check{r})$ is differentiable w.r.t. $\check{r}$ and $\theta$.

*Assumption 2*: the environment and the policies are all deterministic, and the reward $r$ generated by a policy $\pi_\theta$ is differentiable w.r.t. $\theta$ (i.e. $R(s, \pi_\theta(s))$ is differentiable w.r.t. $\theta$).

**Proposition 7.** *If Assumption 1 and 2 hold, the necessary condition for $(\check{r}_k^{\#}, \cdots, \check{r}_K^{\#})$ being an optimal solution to Problem equation $P*$ is, for all $j = k, \cdots, K$,*

$$\frac{\partial \eta(\theta_j)}{\partial \theta_j} \frac{\partial \theta_j}{\partial \check{r}_j}\Big|_{\check{r}_j = \check{r}_j^{\#}} = \sum_{j'=j+1}^{K} \frac{\partial \eta(\theta_{j'})}{\partial \theta_{j'}} \frac{\partial \theta_{j'}}{\partial \theta_{j'-1}} \cdots \frac{\partial \theta_{j+1}}{\partial \theta_j} \frac{\partial \theta_j}{\partial \check{r}_j}\Big|_{\check{r}_j = \check{r}_j^{\#}} \tag{15}$$

*Proof.* Consider the case $K = 2$ for simplicity, and the results naturally extend to a larger $K$.

At iteration 1, Problem equation $P*$ becomes

$$\underset{\check{r}_1, \check{r}_2}{\mathrm{argmin}} \quad \eta(\theta_1) + \eta(\theta_2) \tag{P0}$$
$$\text{s.t.} \quad \theta_1 = f(\theta_0, \check{r}_1)$$
$$\theta_2 = f(\theta_1, \check{r}_2)$$
$$\|\check{r}_1 - r_1\| \leq \epsilon$$
$$\|\check{r}_2 - r_2\| \leq \epsilon$$

where $\theta_0$ and $r_1$ are known.

The relaxed problems are

$$\underset{\check{r}_1}{\mathrm{argmin}} \quad \eta(\theta_1) \tag{P1}$$
$$\text{s.t.} \quad \theta_1 = f(\theta_0, \check{r}_1)$$
$$\|\check{r}_1 - r_1\| \leq \epsilon$$

where $\theta_0$ and $r_1$ are known, and

$$\underset{\check{r}_2}{\mathrm{argmin}} \quad \eta(\theta_2) \tag{P2}$$
$$\text{s.t.} \quad \theta_2 = f(\theta_1, \check{r}_2)$$
$$\|\check{r}_2 - r_2\| \leq \epsilon$$

where $\theta_1$ and $\check{r}_2$ are determined by $\check{r}_1$, the solution to equation $P1$.

Suppose $\check{r}_1^{\#}$ and $\check{r}_2^{\#}$ are the optimal solutions to equation $P1$ and equation $P2$. And we discuss the necessary condition for $\check{r}_1^{\#}$ and $\check{r}_2^{\#}$ being optimal to equation $P0$.

We can rewrite equation $P1$ as

$$\underset{\check{r}_1}{\mathrm{argmin}} \quad \eta(f(\theta_0, \check{r}_1))$$
$$\text{s.t.} \quad \|\check{r}_1 - r_1\|^2 - \epsilon^2 + y_1^2 = 0$$

And the Lagrange function of the above problem is

$$L(\check{r}_1, \lambda_1, y_1) = \eta(f(\theta_0, \check{r}_1)) + \lambda_1(\|\check{r}_1 - r_1\|^2 - \epsilon^2 + y_1^2) \tag{16}$$

The necessary conditions for $\check{r}_1$ being optimal are

$$\frac{\partial \eta(f(\theta_0, \check{r}_1))}{\partial \check{r}_1} + \frac{\partial \lambda_1(\|\check{r}_1 - r_1\|^2 - \epsilon^2 + y_1^2)}{\partial \check{r}_1} = 0 \tag{17}$$

$$\|\check{r}_1 - r_1\|^2 - \epsilon^2 + y_1^2 = 0 \tag{18}$$

$$\lambda_1 y_1 = 0 \tag{19}$$

for some $\lambda_1$ and $y_1$.

Similarly, for Problem equation $P2$, the necessary conditions of optimality are

$$\frac{\partial \eta(f(\theta_1, \check{r}_2))}{\partial \check{r}_2} + \frac{\partial \lambda_2(\|\check{r}_2 - r_2\|^2 - \epsilon^2 + y_2^2)}{\partial \check{r}_2} = 0 \tag{20}$$

$$\|\check{r}_2 - r_2\|^2 - \epsilon^2 + y_2^2 = 0 \tag{21}$$

$$\lambda_2 y_2 = 0 \tag{22}$$

for some $\lambda_2$ and $y_2$.

Since $\check{r}_1^{\#}$ and $\check{r}_2^{\#}$ are the optimal solutions to equation $P1$ and equation $P2$, $\check{r}_1^{\#}$ and $\check{r}_2^{\#}$ satisfy Equation equation 17 $\sim$ equation 22.

Expanding equation 17 and equation 20, we get

$$\frac{\partial \eta}{\partial \theta_1} \frac{\partial \theta_1}{\partial \check{r}_1}\Big|_{\check{r}_1 = \check{r}_1^{\#}} + 2\lambda_1(\check{r}_1^{\#} - r_1) = 0 \tag{23}$$

$$\frac{\partial \eta}{\partial \theta_2} \frac{\partial \theta_2}{\partial \check{r}_2}\Big|_{\check{r}_2 = \check{r}_2^{\#}} + 2\lambda_2(\check{r}_2^{\#} - r_2) = 0 \tag{24}$$

For Problem equation $P0$, the Lagrange is

$$L(\check{r}_1, \check{r}_2 \lambda_1', \lambda_2', y_1', y_2') = \eta(f(\theta_0, \check{r}_1)) + \lambda_1'(\|\check{r}_1 - r_1\|^2 - \epsilon^2 + (y_1')^2) + \eta(f(\theta_2, \check{r}_2)) + \lambda_2'(\|\check{r}_2 - r_2\|^2 - \epsilon^2 + (y_2')^2) \tag{25}$$

And the necessary conditions for $\check{r}_1, \check{r}_2$ being optimal are

$$\frac{\partial \eta(f(\theta_0, \check{r}_1))}{\partial \check{r}_1} + \frac{\partial \eta(f(\theta_1, \check{r}_2))}{\partial \check{r}_1} + \frac{\partial \lambda_1'(\|\check{r}_1 - r_1\|^2 - \epsilon^2 + (y_1')^2)}{\partial \check{r}_1} + \frac{\partial \lambda_2'(\|\check{r}_2 - r_2\|^2 - \epsilon^2 + (y_2')^2)}{\partial \check{r}_1} = 0 \tag{26}$$

$$\frac{\partial \eta(f(\theta_1, \check{r}_2))}{\partial \check{r}_2} + \frac{\partial \lambda_2'(\|\check{r}_2 - r_2\|^2 - \epsilon^2 + (y_2')^2)}{\partial \check{r}_2} = 0 \tag{27}$$

$$\|\check{r}_1 - r_1\|^2 - \epsilon^2 + (y_1')^2 = 0 \tag{28}$$

$$\|\check{r}_2 - r_2\|^2 - \epsilon^2 + (y_2')^2 = 0 \tag{29}$$

$$\lambda_1' y_1' = 0 \tag{30}$$

$$\lambda_2' y_2' = 0 \tag{31}$$

for some $\lambda_1', \lambda_2', y_1', y_2'$ (not the same with $\lambda_1, \lambda_2, y_1, y_2$).

Expanding equation 26 and equation 27, we get

$$\frac{\partial \eta}{\partial \theta_1} \frac{\partial \theta_1}{\partial \check{r}_1}\Big|_{\check{r}_1 = \check{r}_1^{\#}} + \frac{\partial \eta}{\partial \theta_2}\left(\frac{\partial \theta_2}{\partial \theta_1} \frac{\partial \theta_1}{\partial \check{r}_1} + \frac{\partial \theta_2}{\partial \check{r}_2} \frac{\partial \check{r}_2}{\theta_1}\right) \frac{\partial \theta_1}{\partial \check{r}_1}\Big|_{\check{r}_1 = \check{r}_1^{\#}} + 2\lambda_1'(\check{r}_1^{\#} - r_1) + 2\lambda_2'(\check{r}_2^{\#} - r_2)\frac{\partial \check{r}_2}{\partial \theta_1} \frac{\partial \theta_1}{\partial \check{r}_1}\Big|_{\check{r}_1 = \check{r}_1^{\#}} = 0 \tag{32}$$

$$\frac{\partial \eta}{\partial \theta_2} \frac{\partial \theta_2}{\partial \check{r}_2}\Big|_{\check{r}_2 = \check{r}_2^{\#}} + 2\lambda_2'(\check{r}_2^{\#} - r_2) = 0 \tag{33}$$

Combining equation 23, equation 24, equation 32 and equation 33, we obtain

$$\frac{\partial \eta}{\partial \theta_2} \frac{\partial \theta_2}{\partial \theta_1} \frac{\partial \theta_1}{\partial \check{r}_1}\Big|_{\check{r}_1 = \check{r}_1^{\#}} = \zeta \frac{\partial \eta}{\partial \theta_1} \frac{\partial \theta_1}{\partial \check{r}_1}\Big|_{\check{r}_1 = \check{r}_1^{\#}} \tag{34}$$

for some $\zeta$, which is the necessary condition for $\check{r}_1^{\#}$ and $\check{r}_2^{\#}$ being optimal to Problem equation $P0$.

Intuitively, this condition implies that the gradient of $\eta(\theta_1)$ w.r.t. $\check{r}_1^{\#}$ (the RHS) should be aligned with the gradient of $\eta(\theta_2)$ w.r.t. $\check{r}_1^{\#}$ (the LHS), without considering the influence of $\check{r}_1$ to $\check{r}_2$. That is, although $\check{r}_1$ influence $\theta_1$, $\theta_1$ influences both $\theta_2$ and $\check{r}_2$ (because $\check{r}_2$ is generated by $\pi_{\theta_1}$), LHS does not include $\frac{\partial \eta}{\partial \theta_2} \frac{\partial \theta_2}{\partial \check{r}_2} \frac{\partial \check{r}_2}{\partial \theta_1} \frac{\partial \theta_1}{\partial \check{r}_1}$, which makes equation 34 computable in many cases.

However, for the setting of our VA2C-P (monitoring attacker for online RL), the attacker at iteration $k = 1$ does not know the observed reward $r_2$ of iteration $k = 2$, which prevent the agent from conducting the optimal attack. But equation 34 could help the attacker verify whether a past poison is likely to be optimal for the iterations so far. For example, if the learner is using VPG, then at iteration $k = 2$, the attacker can test whether its previous poison $\check{r}_1$ did a good job in minimizing $\eta(\theta_1) + \eta(\theta_2)$ by evaluating whether $(I + \nabla_{\theta_1}^2 \eta(\theta_1)) \nabla_{\theta_2} \eta(\theta_2)$ is equal to $\nabla_{\theta_1} \eta(\theta_1)$.

$\square$

# G EXPERIMENT SETTINGS AND ADDITIONAL RESULTS

## G.1 DETAILED EXPERIMENT SETTINGS

**Network Architecture.** For all the learners, we use a two-layer policy network with $Tanh$ as the activation function, where each layer has 64 nodes. PPO, A2C and ACKTR also have an additional same-sized critic network. We implement VPG and PPO with PyTorch, and the implementation of A2C and ACKTR are modified from the project by Kostrikov (2018).

**Hyper-parameters.** In all experiments, the discount factor $\gamma$ is set to be 0.99. We run VPG and PPO for 1000 episodes on every environment, and update the policy after every episode. For A2C and ACKTR, we use 16 processes to collect observations simultaneously, and update policy every 5 steps (i.e., each observation $\mathcal{O}$ has 80 $(s, a, r)$ tuples); learning last for 80000 steps in total.

**Distance Measure for Perturbation** The definition of total effort function $U(\cdot)$ plays an important role of understanding the attack power. Since states, actions and rewards are in different forms and scales, we define $U$ differently for various poison aims. Also, note that $\mathcal{O}$ is a concatenation of state-action-reward tuples, and its length could vary in different iterations, so we normalize over the length of observation.

For $\mathcal{D} = \mathcal{O}^s$, we define the total effort as

$$U(\mathcal{O}^s, \check{\mathcal{O}}^s) = \frac{1}{\sqrt{|\mathcal{O}^s|}} \sum_{s \in \mathcal{O}^s} \|s - \check{s}\|_2.$$

For $\mathcal{D} = \mathcal{O}^a$, if the action space is continuous, we define the total effort as

$$U(\mathcal{O}^a, \check{\mathcal{O}}^a) = \frac{1}{\sqrt{|\mathcal{O}^a|}} \sum_{a \in \mathcal{O}^a} \|a - \check{a}\|_2,$$

and if the action space is discrete, we define the total effort as

$$U(\mathcal{O}^a, \check{\mathcal{O}}^a) = \frac{1}{\sqrt{|\mathcal{O}^a|}} \sum_{a \in \mathcal{O}^a} \mathbb{1}(a \neq \check{a}).$$

For example, in CartPole, the action is either 0 or 1, then the attacker with $\epsilon = 0.1$ can flip up to 10% of the actions in one iteration.

For $\mathcal{D} = \mathcal{O}^r$, we define the total effort as

$$U(\mathcal{O}^r, \check{\mathcal{O}}^r) = \frac{1}{\sqrt{|\mathcal{O}^r|}} \|\mathbf{r} - \check{\mathbf{r}}\|.$$

In the supplementary materials we provide the code and instructions, as well as demo videos of poisoning A2C in the Hopper environment, where one can see under the same budget constraints, random poisoning has nearly no influence the agent's behaviors, while our proposed VA2C-P successfully prevents the agent from hopping forward.

### G.2 FGSM-BASED POISONING

The procedure of the FGSM-based targeted poisoning is as follows. We transfer the method proposed by Behzadan & Munir (2017) from attacking DQN to attacking policy-based algorithms. Although Behzadan & Munir (2017) assume a black-box setting, we hereby use white-box attack (attacker knows learner's policy parameters) in order to let FGSM be a stronger baseline.

For step $t$,
Step 1: the learner observes $s_t$ takes action $a_t$, the environment returns reward $r_t$, state $s_{t+1}$;
Step 2: the attacker queries the target policy and gets $a_{adv} = \pi^\dagger(s_{t+1})$;
Step 3: the attacker poisons $s_{t+1}$ by

$$\check{s}_{t+1} = s_{t+1} + \epsilon \times \text{sign}(\nabla_{s_{t+1}}(\pi(a_{adv}|s_{t+1})));$$

Step 4: the attacker sends $\check{s}_{t+1}$ as $s_{t+1}$ to the learner.

### G.3 ADDITIONAL EXPERIMENT RESULTS

Figure 8 shows additional results on various environments and RL algorithms, including both non-targeted poisoning and targeted poisoning.

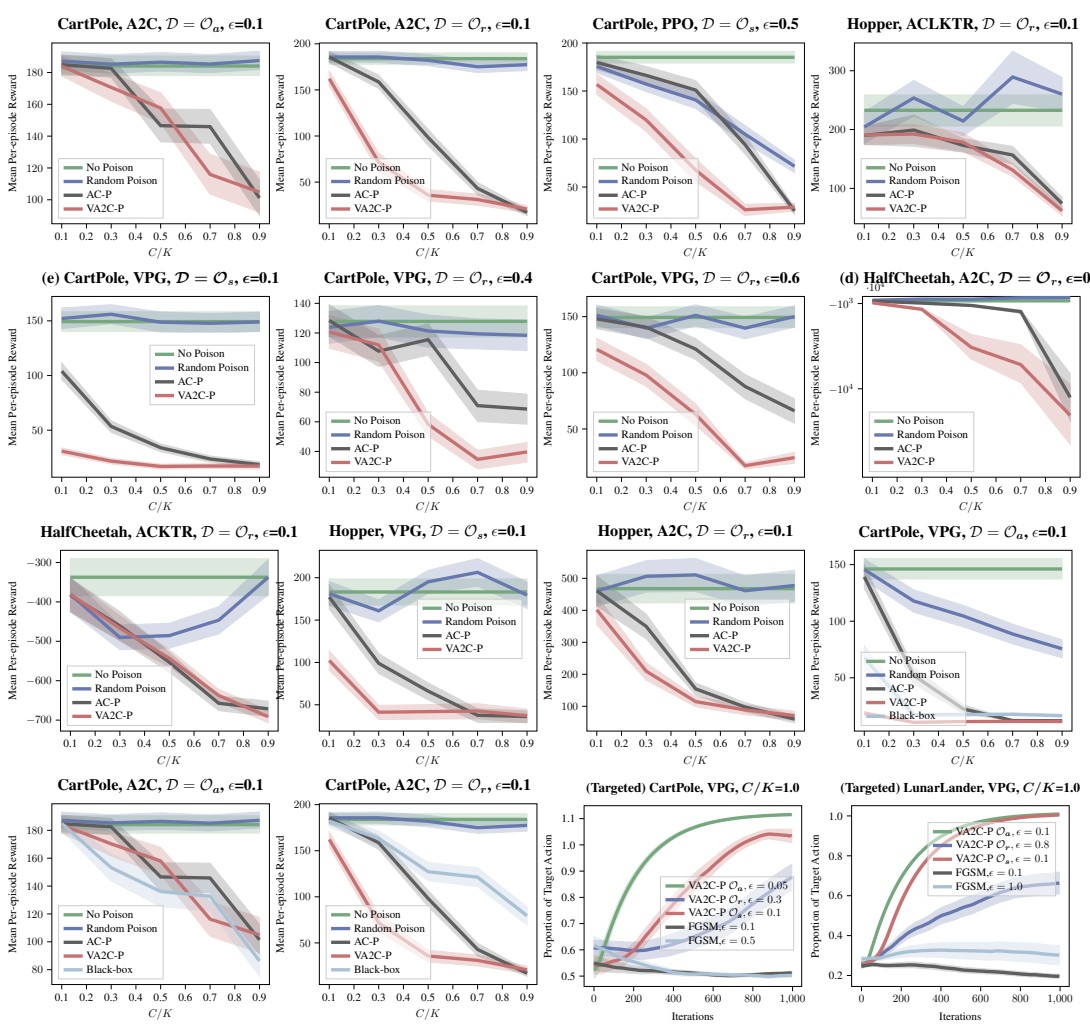

**Figure 8:** Additional Experimental Results

## G.4 EXTENSION TO OFF-POLICY LEARNERS

Although we focus on on-policy policy gradient learners in this paper, our poisoning method is also applicable to off-policy learners which update their policies using sampled mini-batches from all historical observation (trajectories). If the adversary can manipulate the mini-batch the learner samples at every step, our proposed poisoning process works as usual. We implement this idea and test it for one of the state-of-the-art off-policy learning method SAC (Haarnoja et al., 2018), and the results are shown in Figure 9, where VA2C-P significantly reduces the reward gained by the learner. In the other case, if the adversary doesn't see which mini-batch the learner samples but has access to the buffer, he can still alter or insert some samples to influence learning.

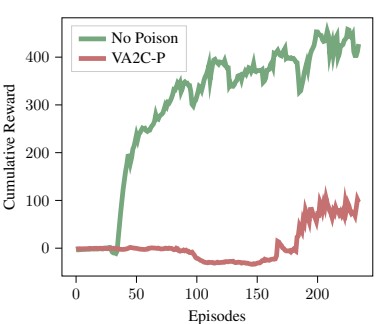

**Figure 9:** Poisoning off-policy algorithm SAC with VA2C-P. $\mathcal{D} = \mathcal{O}^r, C/K = 1, \epsilon = 0.6$.

