# OpenReview forum: "Vulnerability-Aware Poisoning Mechanism for Online RL with Unknown Dynamics"
_ICLR.cc/2021/Conference — ICLR 2021 Poster_

### Official Review · AnonReviewer2 · 2020-10-18
**A good attempt on poisoning RL in unknown environments, but the algorithm is too heuristic and limiting.**

**Rating:** 6
**Confidence:** 4

**Review:**

This paper studies a very important problem of poisoning attack against RL when the attacker is not omniscient. This is an important next step, as most prior work assumes omniscient for the sake of a more rigorous theoretical understanding (e.g. Rakhsha et al., 2020, Zhang et al. 2020).

However, the approach taken by this paper is too heuristic and only applies to a very limited setting where
the learner needs to perform **on-policy** policy gradient methods, which no STOA algorithm does due to its poor sample efficiency, so there isn't much empirical value.

At a high level, the approach this paper takes can be summarized as follows: It defines the optimal poisoning attack problem in an unknown environment as a **sequential decision making problem**, which is well-motivated and clear. It then proposed to simply use a **greedy algorithm** that only optimizes the current step's cost without caring about what happens in the future. Then, of course, the greedy attacker doesn't require the knowledge of the environment's transition. Even if the attacker does know it, it wouldn't be using it anyway, because it only cares about the current step.

And prior work has already shown that in the online data poisoning context, the greedy strategy can be **exponentially worse** than the optimal attack strategy. See [1].

Some technical questions:

1. Correctness of Proposition 2: In the Remarks of section 4.1. It is mentioned that the value difference of two policies differed by at most delta in total variance distance will also be bounded by $poly(\delta, 1/(1-\gamma))$. I didn't check the proof thoroughly but I feel that there are counter-examples?

Consider the classic "combination lock" MDP, where states form a chain of length H, and there are two actions in each state: the "right" action moves you right to the adjacent state, and the "left" action teleports you back to the starting state (the left-most state). All rewards are zero except for when the agent successfully arrives at the right-most state, which takes H right actions consecutively. Now, the optimal policy (always go right) will have value $1$. But a $\delta$-perturbed policy which now has $\delta$ probability of going left will only have value $(1-\delta)^H$ (assuming a fixed episode length H and no discounting). So the gap seems to be exponential rather than polynomial. How does this example fit into the conclusion of proposition 2?

2. About adversarial critic: What exactly is this $\tilde V_\omega$ estimating? Is it estimating the value of the optimal policy, the learner's current behavior policy, or the learner's current $\hat\pi_k$ which according to the paper is different from the behavior policy? If it was the latter two, how does the attacker keep track of the value of an ever-changing policy, and how does he use this value estimate to evaluate the value of some other potential poisoned policy? It's all very confusing.

3. The estimated rank in step 6 of section 4.3 is only unbiased if the policy discrepancies $\hat \psi_k$ are *i.i.d.*, which they clearly are not.



[1] Xuezhou Zhang, Xiaojin Zhu, Laurent Lessard. Online Data Poisoning Attack. L4DC 2020.

---

> ### Author Response · Authors · 2020-11-16
> **[Response to R#2 - Part 3] R#2's example does not violate Proposition 2; an explanation of the adversarial critic**
>
> >**Q3: Does the combination lock MDP violate Proposition 2?**
>
> A3: This example **does not violate our Prop. 2**, because Prop. 2 upper bounds the **reward/value drop** caused by poisoning, i.e., $V_{\pi}(s_0)-V_{\pi^\prime}(s_0)$. In the combination lock MDP, if we assume non-discounting($\gamma=1$) as R#2 assumes, then our upper bound becomes $\infty$ since $(1-\gamma)$ is in the denominator, thus the upper bound is valid; in the other case, if $0\leq\gamma<1$, the inequality given by Prop. 2 is
> $$
> V_{\pi}(s_0)-V_{\pi^\prime}(s_0) = \gamma^H - (1-\delta)^H \gamma^H \leq  a\frac{\delta^2\gamma}{(1-\gamma)^2} + b\delta
> $$
> where $a$ and $b$ are independent of $\delta,\gamma$ and $\pi^\prime$, as detailed by our Prop. 2 in Appendix D.
> Although it is hard to derive an analytical proof of the inequality due to the complex form, we implemented a numerical grid test for \delta=0:0.001:1, \gamma=0:0.001:1, and for different H’s, none of them violates the above inequality.
>
>
> ---
> >**Q4: What exactly is the adversarial critic $\tilde{V}_\omega$ estimating? how does the attacker keep track of the value of an ever-changing policy, and how does he use this value estimate to evaluate the value of some other potential poisoned policy?**
>
> A4:
> (1) **What is the adversarial critic estimating?**
> $\tilde{V}_\omega$ is the value function of the learner's current behavior policy (as we stated in Section 4.2, it is trained with the unpoisoned trajectories generated by the learner's policy). In other words, $\tilde{V}_\omega$ is the critic for the learner's current policy. Note that the critic/value function learned by the learner is not the correct value for its policy, because the learner uses poisoned trajectories to update its critic. And the attacker, on the contrary, observes the clean trajectories generated by the learner's policy, thus can learn a correct critic for the policy.
>
>
> (2) **How does the attacker keep track of the value of an ever-changing policy, and how does he use this value estimate to evaluate the value of some other potential poisoned policy?**
> $\tilde{V}_\omega$ is the value of the policy in the current iteration, and the value will be updated in the next iteration when the policy changes. It is exactly the Actor-Critic framework, but the critic is owned by the adversary.
> For estimating the value of a potential poisoned policy, we use importance sampling, as we described in the last two paragraphs in Section 4.2.
>
> We admit that in the original submission, some messages about the adversarial critic are embedded in the descriptions and might be hard to find for readers, although we did mention them. We have modified the paper to make these details more explicit.
>
>
> ---
> >**Q5: The estimated rank in step 6 of section 4.3 is not unbiased.**
>
> A5: R#2 is right that the estimated rank is not unbiased. We apologize for this imprecise description. But the algorithm itself does not need the rank to be unbiased, we choose this rank estimation because it works much better than simply ranking the historical policy discrepancies. We have removed the word "unbiased" in our modified version.

---

> > ### Comment · AnonReviewer2 · 2020-11-16
> > **Follow up on Q3**
> >
> > OK, so if we are talking about discounted MDPs, here is another example:
> > there are two states and two actions, $a_1$ transit $s_1$ to $s_1$ with reward 1, and $a_2$ transit $s_1$ to $s_2$ with reward 0, and $s_2$ is an absorbing state, with both actions transit to itself with reward 0. Suppose that the agent starts in $s_1$, and $\pi$ performs action $a_1$ with probability 1. So $V^\pi(s_1) = \frac{1}{1-\gamma}$. $\pi'$ performs action $a_2$ with probability $\delta$, so $V^\pi(s_1) = \frac{1}{1-\gamma(1-\delta)}$, so their difference is
> > $$
> > V^\pi(s_1) - V^\pi(s_1) = \frac{1}{1-\gamma} - \frac{1}{1-\gamma(1-\delta)}
> > $$
> > My calculation shows that if $\gamma =0.99$, and $\delta=0.01$, then $x = \frac{1}{1-\gamma} - \frac{1}{1-\gamma(1-\delta)}\approx 49.75$, whereas the upperbound $y = \frac{\delta^2\gamma}{(1-\gamma)^2}+\delta \leq 1$. If $\delta$ continues to decrease, the ratio $x/y$ will continue to increase. What are the constant $a$ and $b$ here in this problem instance?

---

> > > ### Author Response · Authors · 2020-11-17
> > > **Explanation for this new example**
> > >
> > > We would like to point out in our Proposition 2, $a$ and $b$ are dependent on the advantage function of the un-poisoned policy $\pi$. (In our last response, we only said that $a,b$ do not depend on $\delta,\gamma$ and $\pi^\prime$). More specifically, the upper bound given by Proposition 2 is
> > >
> > > $\frac{4 \delta^2 \gamma \max_{s, a} | A_{\pi}(s, a)|}{(1-\gamma)^{2}}+2\delta E_{s\sim\pi}[\max_a A_{\pi}(s,a)]-E_{s,a\sim\pi}[A_\pi(s,a)]$
> > >
> > > In the example given by R#2, we have $\max_{s, a} \left| A_{\pi}(s, a)\right|=|Q_{\pi}(s_1,a_2)-V_{\pi}(s_1)|= \frac{1}{1-\gamma}$, the second and the third term are zeros, so the upper bound is still larger than the value difference. Therefore, **Proposition 2 is still valid**.
> > >
> > >
> > > In Remark (3) under Definition 1, we say the upper bound for value drop is $O\left(\delta^{2} \gamma(1-\gamma)^{-2}\right)$ because the clean next-policy and its advantage function are already fixed. But the bound is not only determined by $\delta$ and $\gamma$. The purpose of Remark (3) is just to provide some insights on how the value drop depends on the policy discrepancy, and we put the formal bound in Proposition 2 in appendix. R#2's question reminds us that the remark itself may confuse some readers if they do not see the complete expression in the appendix, so we will explain it in a revised version. Thanks!

---

> > > > ### Comment · AnonReviewer2 · 2020-11-17
> > > > **Followup still**
> > > >
> > > > OK, but if $\delta=0.001$, $\gamma=0.99$, then
> > > > $$
> > > > \frac{4 \delta^{2} \gamma }{(1-\gamma)^{3}}/\left(\frac{1}{1-\gamma}-\frac{1}{1-\gamma(1-\delta)}\right) = 0.44<1,
> > > > $$
> > > > Please do verify my calculations..

---

> > > > > ### Author Response · Authors · 2020-11-17
> > > > > **The second term has a typo, although does not influence the algorithm**
> > > > >
> > > > > We have verified R#2's calculation, and found that the second term we calculated in our previous response was incorrect. The second term is not zero, it should be $2\delta \sum_{s\in S} g_{\pi}(s) \max_a  A_{\pi} |(s,a)|$, where $g_{\pi}(s):=\sum_{t=0}^{\infty} \gamma^t P(s_t=s|\pi)$. In this example, $g_{\pi}(s_1)=\frac{1}{1-\gamma}$, $g_{\pi}(s_2)=0$, and $\max_a |A_{\pi}(s_1,a)|=\frac{1}{1-\gamma}$, therefore, the second term is $\frac{2\delta}{(1-\gamma)^2}$. Then, $LHS=\frac{1}{1-\gamma}-\frac{1}{1-\gamma(1-\delta)}$, $RHS=\frac{4\delta^2\gamma}{(1-\gamma)^3}+\frac{2\delta}{(1-\gamma)^2}$. One can verify that $\frac{LHS}{RHS} \leq 1$.
> > > > >
> > > > > This miscalculation is because of a typo in the second term in our appendix, and we have fixed it. We apologize for the previous mistake, and greatly appreciate R#2's insightful example!
> > > > >
> > > > > Nevertheless, note that the first term is still correct, which dominates the bound in most cases, and the result shown in the main body of the paper does not have the problem, so it will not influence our proposed algorithm.

---

> ### Author Response · Authors · 2020-11-16
> **[Response to R#2 - Part 2] Our method is different from the greedy attack in literature; our work already considers more future than prior works**
>
> >**Q2: Is the greedy algorithm too simple and exponentially worse than an optimal strategy as shown in [2]?**
>
> A2:
>
> (1) Our method is not naively getting the greedy solution at every step. The original problem (Q) is a mixed integer program, which is NP-hard even if the dynamics are known. And we tackle the hard integer constraint by proposing a "**selective greedy**" method, which selects poisoning aims or steps by their vulnerability. We are also the first to propose a principled vulnerability measure with theoretical interpretation. Please note that **existing poisoning methods** for deep RL are more "greedy" (based on simpler heuristics) than ours, since most of them only perturb one state[3], while we perturb the whole trajectory. Also, as mentioned in [Response to R#2 - Part 1], we **consider the next iteration rather than the current one**. In our experiment, we show that our method outperforms existing poisoning methods[3]. So we are already one step closer than existing methods toward "optimality".
>
> (2) Although under the worst scenario, the greedy algorithms could be exponentially worse than optimal strategies, it is not necessarily true under most scenarios. As we show in appendix F, the greedy direction could be the same as the optimal direction, although we admit that the performance of greedy methods is often worse than optimal ones. Also, we emphasize that **our algorithm is different from the greedy algorithm evaluated in [2]** in terms of the "degree of greedy". The greedy baseline in [2] only considers one data sample at one time, but our method chooses the optimal solution for **a batch of samples**. Thus the greedy method in [2] does not work well (has similar results with no poisoning), but our method works very well in experiments (reduces the reward by a lot or quickly leads the agent to a target policy).
> More importantly, there is often a **trade-off between optimality and efficiency**. Greedy algorithms are usually fast, so that they can be used in many applications. Many successful algorithms are based on greedy heuristics. So we do not think being greedy necessarily means bad. Actually, our experiments show that our VA2C-P works effectively and efficiently in a wide variety of scenarios.
>
> (3) People may not use a greedy algorithm only when they have better choices. However, **there is no practical optimal strategy yet for the online deep RL poisoning**, so our VA2C-P is the current best choice. The optimal strategy proposed in [2] can not be applied to our online deep RL poisoning problem, which is intrinsically more challenging than the setting discussed in [2]. [2] formulates the online poisoning problem as an MDP for the adversary, and solves the MDP either by nonlinear programming, or by directly training a DDPG agent with pre-attack data.
> However, building such an adversary MDP is prohibitively difficult in our case due to the reasons listed in **(a)**, **(b)** and **\(c\)** below. **(a)** Paper [2] defines the state space as the product space of victim model $\theta$ and the **current input data sample** $z_t$, while in our case, the input data sample is the product space of victim model $\theta$ and **a set of recent trajectories** $O_t=\{s_1,a_1,r_1, \cdots\}$, making the state space extremely large. **(b)** The reward function of the adversary MDP in [2] is defined as a simple function of $\theta$ (e.g. $||\theta-\theta_{target}||$), so they can directly evaluate the reward of a state and a poisoning action. But in our reward-minimizing case, the reward of the adversary is defined as the value of a policy after poisoning, which requires extra estimating. **\(c\)** More importantly, [2] assumes the training data samples $z_t$ are i.i.d. from a time-invariant distribution, so that the adversary MDP is stationary, and the DDPG agent trained on pre-attack data can work for future steps. However, in our online RL setting, the data samples are not i.i.d. and the data distribution changes with the victim's policy (because the trajectory distribution depends on the behavior policy). Therefore, even if we build a large MDP regardless of the complexity discussed in (a) and (b), the MDP is changing, and a good policy for the current MDP may fail in future rounds.
>
>
> Finally, poisoning online deep RL is a new field. Our paper first formulates this problem for a wide range of deep RL algorithms. As pointed out by R#1, this problem is generally hard and intractable, so it is natural to relax the problem and find approximate solutions. We admit that our proposed algorithm is not the theoretically optimal solution to the original NP-hard problem, but it partially solves an important part of the problem, and works well in practice. We also provide some theoretical analysis for the relaxation in Appendix F.
>
> ---
> Refs:
>
> [2] Zhang, et al. Online Data Poisoning Attack.
>
> [3] Behzadan, et al. Vulnerability of deep reinforcement learning to policy induction attacks.

---

> > ### Author Response · Authors · 2020-11-16
> > **[Response to R#2 - Part 2] -- Continued**
> >
> > In summary, we believe our paper takes an **important and adequately large step in this new direction**. And we hope that reviewers could understand the challenges embedded in this problem, and consider our contributions to the existing literature which lacks even a nonoptimal but practical solution for the original challenging problem.

---

> > ### Comment · AnonReviewer2 · 2020-11-17
> > **Updated review**
> >
> > Thanks for the detailed explanations. The rebuttal addressed most of my concerns. I will therefore increase my score to 6.

---

> > > ### Author Response · Authors · 2020-11-17
> > > **Thank you!**
> > >
> > > Thank you so much for your encouragement! Your feedback has helped us a lot.

---

> ### Author Response · Authors · 2020-11-16
> **[Response to R#2 - Part 1] Clarification of the misunderstanding; on-policy policy gradients are useful**
>
> We appreciate the valuable feedback from R#2, and we address the major concerns R#2 raises as follows.
>
> ---
> We want to first point out that R#2 may have some misunderstanding of our problem formulation and algorithm.
> We notice this sentence by R#2: *"Then, of course, the greedy attacker doesn't require the knowledge of the environment's transition. Even if the attacker does know it, it wouldn't be using it anyway, because it only cares about the current step."*
> We would like to clarify this misunderstanding: the knowledge of the environment is very important even if only considering the current step, because **the current step attack is to minimize the loss of the next step**, which requires prediction or estimation. As the objective of the relaxed Problem \(P\) shows, it is the attacker's loss for $\pi_{j+1}$, the **next-step poisoned policy**. So **we do consider the future**.
>
> Now we answer the questions R#2 has in this comment and the following two comments, including the concern about **on-policy policy gradient methods**, **the greedy algorithm**, and some **technical questions**
>
> ---
> >**Q1. Are on-policy methods empirically useless? Can our method work on off-policy algorithms?**
>
> A1. (1) On-policy policy gradient methods are still important and widely used nowadays. The methods we use in our experiments, such as A2C, PPO and ACKTR, are all successfully and widely used in deep RL applications and research, incorporated in most modern deep RL libraries[1]. Off-policy methods and on-policy methods both have pros and cons, as the former are more sample efficient, but the latter have better convergence behavior and are more stable. So we respectfully disagree with R#2's opinion that poisoning on-policy learners is not important. Our paper proposes the **first attempt to poison deep policy gradient methods**, covering all on-policy RL methods. R#1, R#2 and R#4 all agree that our work has notable empirical significance compared with prior works.
>
> (2) **Our poisoning algorithm can be used to poison off-policy victims**:
> (a) if the adversary can manipulate the minibatch the learner samples at every step, our proposed poisoning process works as usual. (We tested it on SAC, the mean reward is 312.6 for non-poison vs 11.1 under our poison. We will report the results in our modified version.)
> (b) otherwise, if the adversary doesn’t see which minibatch the learner samples but has access to the buffer, he can still alter or insert some samples to influence learning.
>
> ---
> Refs:
>
> [1] Hill, et al. Stable Baselines. 2018.

---

### Official Review · AnonReviewer1 · 2020-10-27
**Review for Vulnerability-Aware Poisoning Mechanism for Online RL with Unknown Dynamics**

**Rating:** 7
**Confidence:** 5

**Review:**

The paper studies poisoning attacks against online reinforcement learning agents. The attacker has the power of manipulating the training data, i.e., state-action-reward trajectories, in order to achieve some attack goal. The attack can be completely black-box, meaning that the proposed method allows an attack setting where the attacker has no knowledge of the RL algorithm used by the victim agent or the environment. In this scenario, the authors proposed that the attacker can imitate the learning procedure of the victim, and then based on the imitated policy; the attacker designs how to poison the training data. The attack is formulated as a bi-level optimization, where the lower level involves the imitated learning procedure. Due to the intractability of sequential optimization, the original formulation is simplified so that only the attack only solves the attack on the current training data. This procedure is repeated in every episode to achieve sequential attacks. Experiments on a variety of tasks demonstrate the superiority of the proposed attack.

Compared to prior works, the main advantage of this paper lies in that the attack can be applied in more complicated tasks where state or action space is continuous. Furthermore, the attack takes into account the adversarial effect of the current attack on future behavior of the victim agent. Therefore, the attack achieves better overall performance (e.g. more times of target-action selection) than traditional gradient-based attack such as FGSM.

Another strength of the paper is that it provides some theoretical analysis in terms of how the relaxed optimization approximates the original complex attack optimization. This is in general a hard question to answer. Although the analysis is only about attack feasibility and how to test sub-optimality in hindsight, there is value in deriving those theoretical results.

The experimental part of the paper is also strong. The authors have shown convincing results that demonstrate the proposed VA2C-P attack outperforms existing gradient-based FGSM attacks. Moreover, there are systematic empirical investigations on how the attack constraint parameter epsilon affects the attack performance.

Finally the paper is well-written and provides a nice summary of prior works, as well as why each prior work fails to achieve some desired property of attack in an ideal sequential attack scenario. Therefore, overall I think the paper is nice and makes significant contribution.

One disadvantage of the paper is that while the paper claims able to handle sequential attacks, the relaxed attack optimization seems solving only the desired manipulation on training data in the current episode. As a result, the solution is definitely sub-optimal. The authors provided a method of evaluating whether the implemented attack is sub-optimal in hindsight, specifically the proposition 7. This result, while being interesting, is not useful in that it cannot help practitioners gauge if the computed attack is optimal or not, since it’s a necessary condition. I am wondering if the authors could provide some insight on how sub-optimal is the attack in this paper, and potential ways to further improve it.

I also want to point out that the attacker knowledge assumed in this paper is not strictly less than prior works such as in Zhang 2020. Both need to have access to a simulator of the MDP and victim environment, and both need to exhaust large amount of training data before obtaining a good attack policy. In this paper, the attacker needs to imitate the victim learner, and the accuracy of imitation result will depend on how many training data are available.

---

> ### Author Response · Authors · 2020-11-16
> **[Response to R#1] Measuring sub-optimality, potential ways to improve, and comparison with related works**
>
> We appreciate the positive feedback and constructive suggestions provided by R#1. Especially, we appreciate a lot that R#1 noticed our theoretical analysis about the relaxation and thought them valuable.
>
>
> We address some questions raised by R#1 as follows:
>
> ---
> >**Q1: How to determine the sub-optimality of the attack?**
>
> A1: (Insights on measuring sub-optimality)
>
> R#1 is right that Proposition 7 provides a necessary condition for the attack being optimal. And we admit that due to the high non-convexity, it is difficult to judge whether an attack is globally optimal for the whole attacking process. However, Prop. 7 also provides an idea of judging how likely a past attack decision is locally optimal. Equation (15) essentially measures whether the actual search (attack) direction matches the optimal direction. If they match, then the attack is at least locally optimal; if they do not match, then the attack is nonoptimal even locally. In practice, we want the LHS of Equation (15) to be as **aligned to the RHS as possible** for local optimality, because it suggests that the attacker is leading the agent to the desired direction. Otherwise, if the search direction is opposite to the optimal direction, one may consider changing the attack strategy (but note that being nonoptimal does not necessarily mean that the current poisoning is bad; it may still deprave the victim significantly).
>
>
> ---
> >**Q2: Are there ways to improve the optimality of attacking?**
>
> A2:
> There are some potential ways to further improve our proposed attack, although they may require more computations and knowledge.
>
> (1) The attacker can fit a prediction model for the unknown environment (predict future states and rewards given the current state and action) using the trajectories generated by the victim policy. Then the attacker can predict future trajectories. In this case, we can use Proposition 7 as "foresight" rather than "hindsight". More specifically, we may look $N$-iterations ahead, and figure out a poison direction that is optimal for the next $N$ iterations instead of the next one iteration. However, it may require more computations than our current method.
>
> (2) If the attacker is allowed to directly interact with the environment (which is not allowed in our paper), or even the attacker knows the dynamics of the environment, then it may be possible to pre-compute the "ideal" attacks. For example, assume one is doing targeted poisoning, then using Inverse RL, one can design an "ideal" reward function such that any agent will learn the target policy if the rewards are given by the "ideal" reward function. (Although Inverse RL focuses on teaching the agent a good policy, the adversary can do the opposite and set a malicious policy as the target.) Then during the victim's online learning process, the goal of the attacker is to perturb the actual observations toward the "ideal" attack directions as much as possible under their budget and power constraints. This simplified problem is still hard to solve due to the limited budget and power, but it would be more effective since it has extra knowledge of the environment.
>
>
> ---
> >**Q3: Is the required knowledge less than prior work?**
>
>
> A3:
> As mentioned in A2(2), we would like to clarify that our proposed attacker **needs not to interact with the MDP or a simulator**; we only use the learner's observations generated via the learner's interaction with the environment.
>
> The word "imitate" might be misleading to readers, and we will make it more clear in the modified version. In our Problem (Q) line (b), the notation $\mathcal{O}_j$ is the observation that is already gained by the learner/victim in the $j$-th iteration, and $\check{\mathcal{O}}_j$ denotes the perturbed/poisoned observation with a specific poison aim $\check{\mathcal{D}}_j$ ($\check{\mathcal{D}}_j$ is to be determined by the attacker himself). And imitating just means that the attacker wants to follow what the victim would do once it receives the poisoned observation $\check{ \mathcal{O} }_j$. In other words, the attacker maintains its own "copy of victim" by feeding it with whatever training data the actual victim uses. Therefore, the attacker only needs to "eavesdrop" on the interactions between the learner and the environment, and requires **no extra training data from the environment**.
>
> In contrast, paper[1] mentioned by R#1 proposes a white-box reward-poisoning method, as well as sound theoretical results for the feasibility and optimality of targeted attacks, although [1] focuses on finite MDPs and the attacker requires the knowledge of the MDP parameters. In contrast, our attacking strategy does not require knowledge of the MDP, and works for large, continuous MDPs, although optimality is not guaranteed. Therefore, our poisoning method and the method in [1] work in different scenarios.
>
>
> ---
> Refs:
>
> [1] Zhang, et al. Adaptive Reward-Poisoning Attacks against Reinforcement Learning. 2020.

---

### Official Review · AnonReviewer3 · 2020-10-28
**A novel poisoning method against policy-based RL agents**

**Rating:** 6
**Confidence:** 3

**Review:**

Summary:
This paper proposes a poisoning algorithm named Vulnerability-Aware Adversarial Critic Poison (VA2C-P) to attack policy-based deep reinforcement learning agents. The poisoning attack is formulated as a sequential bilevel optimisation problem (Problem Q), where the attacker either minimises the expected total rewards of the learner (non-targeted poisoning), or forces the learner to learn a target policy (targeted poisoning). To solve Problem Q, VA2C-P mainly makes two decision: (1) when to attack: a new metric named stability radius is proposed to decide the attack timing, (2) how to attack: a mechanism of adversarial critic is designed to solve a relaxed version of Problem Q by only considering the loss of the immediate next iteration.

Pros:
1. It is an important question to investigate how policy-based RL algorithms can be poisoned by adversarial attacks.

2. It is novel to propose a poisoning method against policy-based RL agents, which has not been studied before.

3. The proposed poisoning framework (Problem Q) is a general formulation that covers a variety of models.

4. VA2C-P has been demonstrated to be effective in targeted and untargeted attacks, under both white-box and black-box settings.

Cons:
1. This paper considers a scenario where (1) “the attacker does not know the underlying dynamics of MDP, and cannot directly interact with the environment, either”; (2) the attacker is able to obtain the states observed by the agent, their actions and rewards. Is (2) a realistic setting? Especially, how is the reward accessible to the attacker?

2. The black-box attack studied in the paper is closer to white-box attack than to black-box attack: the attacker can still access the past and current observations (state + action + reward), and the only limit is that the policy $\pi$ of the target model is unknown. This type of black-box attack is unrealistic in many situations.

---

> ### Author Response · Authors · 2020-11-16
> **[Response to R#3] Explanation about the attacker's knowledge and comparison with related works**
>
> We thank R#3 for the detailed summarization and valuable comments. And we address the concerns R#3 mentioned as follows, most of which are related to the knowledge of a poisoning attacker.
>
>
> ---
>
> >**Q1: This paper considers a scenario where (1) “the attacker does not know the underlying dynamics of MDP, and cannot directly interact with the environment, either”; (2) the attacker is able to obtain the states observed by the agent, their actions and rewards. Is (2) a realistic setting? Especially, how is the reward accessible to the attacker?**
>
> A1:
> In many RL applications, an agent learns by interacting with the outside environment. For example, learning how to drive on a road, learning how to recommend on a website, learning how to communicate by talking with people online, etc. In these cases, an attacker may eavesdrop on the interactions and alter them.
>
> Especially, the reward is usually a signal sent from the environment to the agent. For instance, consider a chat robot that learns by talking with people via the internet. The reward can be simply defined as the rating score people give to it after the chat. The rating scores are submitted by people from their devices, and transmitted through the internet into the robot's server. In this process, an attacker can perform man-in-the-middle attacks, blocking and changing the scores.
>
> Our Figure 5 in Appendix B visualizes this process and compares it with the poisoning process in supervised learning. Essentially, poisoning is to alter the training data, and in RL, the "training data" is just the interaction trajectories (states, actions and rewards). Thus the access to states, actions and rewards is analogous to the access to the training data, which is a common assumption in papers about poisoning.
>
> In addition, we would like to point out that almost all existing RL poisoning works[1-4] assume access to the interaction data (states, actions and rewards), and most of them[1,3,4] also assume knowledge of to the MDP dynamics, or assume the attacker can directly interact with the environment[2].
>
>
>
> ---
>
> >**Q2: The black-box attack studied in the paper is closer to white-box attack than to black-box attack: the attacker can still access the past and current observations (state + action + reward), and the only limit is that the policy $\pi$ of the target model is unknown. This type of black-box attack is unrealistic in many situations.**
>
> A2:
> To the best of our knowledge, black-box attacking mainly refers to the case where the attacker does not know the learner's model. Many popular works on black-box attacking[5] assume that the attacker is able to "observe labels assigned by the DNN for chosen inputs", which is similar to our setting in the RL regime, since states are the inputs to the DNN, actions are the outputs.
>
> However, we agree that investigating the attacks under the scenarios that the attacker knows even less, especially with no knowledge of the reward, is also an important research topic. We would like to explore how a black-box attacker with knowledge of neither the learner's model nor the learner's reward history could poison the agent in the future.
>
>
>
> ---
> Refs:
>
> [1] Amin Rakhsha, et al. Policy teaching via environment poisoning: Training-time adversarial attacks against reinforcement learning.
>
> [2] Vahid Behzadan, et al. Vulnerability of deep reinforcement learning to policy induction attacks.
>
> [3] Yunhan Huang, et al. Deceptive Reinforcement Learning Under Adversarial Manipulations on Cost Signals.
>
> [4] Yuzhe Ma, et al. Policy poisoning in batch reinforcement learning and control.
>
> [5] Nicolas Papernot, et al. Practical Black-Box Attacks against Machine Learning

---

### Official Review · AnonReviewer4 · 2020-10-28
**Official Blind Review #4**

**Rating:** 6
**Confidence:** 3

**Review:**

#### Summary:
The paper studies poisoning attacks on RL agents, in which the attacker influences the agent's learning process by changing the feedback obtained from the environment. The focus is put on attacking policy-based deep RL agents, without necessarily having access to the underlying MDP model of the environment. The paper proposes a new poisoning algorithm, called Vulnerability-Aware Adversarial Critic Poison, and experimentally demonstrates its effectiveness on 5 different RL environments.

#### High level comments (pros & cons):

-The paper studies an important and interesting topic, poisoning attacks on RL agents, and develops a novel deep learning methods for designing more efficient and scalable attack strategies.

-In contrast to prior work, the proposed algorithm utilizes deep RL techniques, making it applicable to more complex environments. The experimental results indicate the usefulness of the proposed method.

-The clarity of the paper could be improved, including the statement and description of the optimization problem and the algorithm. Furthermore, it is not clear how the problem formulation compares to prior work. Some claims in the paper should be stated more precisely.

-The proposed algorithm is a greedy approach and does not have provable guarantees on the optimality of the derived attack strategies. This is in contrast to prior work cited in this paper, which appears to have some guarantees on the performance of the attack.

Overall, I enjoyed reading the paper, and its contributions seem novel and important for the line of work on poisoning attacks in RL. However, I also think that the paper could be improved in terms of clarity, and additional justifications and explanations could be added throughout the paper.

#### More specific comments and suggestions for improvement:

-The exposition of the results could be improved, and some parts clarified and made more precise. For example, footnote 1 is confusing, since it states that this paper assumes that the attacker poisons observations. On the other hand, the paper also mentions results on 'Hybrid aim poisoning'. It is also confusing that optimization problem (Q) is defined as weighted loss, and then in the first paragraph of Section 4 we have the claim: 'Without loss of generality, we assume the loss weights $j = 1$ for all $j = 1,..., K$.'. The first sentence in Challenge II is also not clear: why is Markovian property important in 'are no longer i.i.d. due to the Markovian property'?  There are also sentences that do not seem to be precise. E.g., the sentence 'However, in complex environments such as Atari games, knowing the dynamics of the MDP is difficult.'  doesn't seem to be precise (since dynamics can be obtained from Atari simulator...).  Given that the paper motivates its setting with Atari games, it is also not clear why Atari games were not used as a test-bed.

-Parts of the optimization problem (Q) are somewhat confusing. In particular, the paragraph that explains constraint (b) (imitate the learner), does not seem to precisely specify what this constraint looks like. In the white-box attack, it is written that the attacker knows the learner's policy and can directly copy it.  It is also stated for black-box attacker that it 'may know the learner’s algorithm', but does not know the learner's policy. On the other hand, in Section 2, it is written that 'white-box attackers, who know the learner’s model, and black-box attackers, who do not know the learner’s model. '.  These parts could be explained in more detail, or more precisely stated.

-Constraints (c) and (d) control the attacker's influence, but the paper does not seem to indicate the practical importance of having both constraints. It might be useful add some discussion on this, as having both constraints seems to affects the algorithmic design proposed in the paper, and the complexity of the optimization problem.

-The discussion in the related work section seems to put emphasis on practical importance of the proposed approach compared to some of the recent papers on poisoning attacks. However, it does not seem to elaborate on the differences in problem formulations, i.e., optimization problems and objectives.

-The focus seems to be on an episodic setting in which after each episode (iteration), an RL agents updates its policy after the data is possibly poisoned. I'm wondering to what extent would these result generalize to fully online setting in which an agent can change its policy after each action taken. Moreover, it is not clear how one can poison e.g. observations only after an episode ends (since the same observations are needed to derive actions from the agent's policy). Additional discussion on this would be valuable.

-The model of the black box-attack is somewhat ambiguous. It is first stated that a black-box attacker may know the learner's algorithm, but it is not specified to what extent the attacker relies on the knowledge about the learner's algorithm. Furthermore, the notion of pseudo-learner does not seem to be defined.

-Minor: There are two 'Step 5' in the description of the algorithm. Furthermore, the algorithm uses variables $\psi$ and $\Psi$, which do not seem to be defined before section 4.3.

The paper also contains minor typos, e.g.:
- the test-time evasion attacks Chen et al. (2019) where the attacker crafts -> citation style
-alter the environment (e.g. change the transition probabilities) ,  -> remove space before ,
-On the contrary, We consider non-omniscient attackers - 'We' should be 'we'
-'a learner gains from the environment, i.e., $O = (O^s,O^a,O^r)$' -> Is $O^a$ observed?
-Compared with Problem ( Q), -> ( Q) should be (Q)
-The solution to Problem (P) is always feasible to Problem (Q, although... -> e.g. (Q should be (Q)
etc.


#### Questions:

a. I didn't understand footnote 1. It states that this paper assumes that the attacker poisons observations, but on the other hand, the results seem to suggest that other attacks are also considered. Could you clarify what types of attacks are considered in the paper?

b. The optimization problem (Q) seems to be different from the ones studied in (Ma et al. 2019) and (Rakhsha et al. 2020). Could you elaborate on the differences between these attack formulations? How does your setting relate to the setting of (Zhang and Parkes, 2008) in terms of computational complexity?

c. The budget constraint (c) in optimization problem (Q) assumes that the 'cost' of an attack is either 0 or 1, whereas the constraint (d) already limits the 'power' of the attack. Could you explain the practical importance of imposing these two constraints together?

d. Which practical application would support the episodic poisoning setting described in this paper?

e. How exactly is the pseudo-learner defined in this paper (e.g., in Section 5)?

---

> ### Author Response · Authors · 2020-11-16
> **Summarization of Our Response**
>
> We thank R#4 for the valuable and detailed feedback.
>
> Most of R#4's concerns are on the clarity of demonstrations and detailed explanations of the method. Due to the complexity of poisoning RL, lots of new and notations/concepts are required in our paper. For clarity, we organized the most important messages/concepts and provided as many intuitive explanations as possible through the main paper; therefore the reader might have to find clarifications from the appendix for some technical details. For example, the **detailed poisoning formulation and comparison with related works** are in Appendix B; more rigorous definitions of the stability radius are in Appendix D; **detailed algorithm illustration** is in Appendix E; we also provide some **theoretical analysis** on the relaxation problem in Appendix F.
>
> We understand that reviewers are not obligated to find all details in the appendix, therefore we greatly appreciate the reviewer's comments, and have modified the main body of the paper according to R#4's valuable suggestions.
>
>
> We summarize all questions raised by R#4 into 4 categories.
>
> **Part 1. Questions and confusion due to some misinterpretation of our paper.**
>
> **Part 2. Questions about problem formulation and theory.**
>
> **Part 3. Questions about algorithm details and extensions.**
>
> **Part 4. Questions due to imprecise/ambiguous wording in our previous manuscript.**
>
> We will address all questions in each category respectively in 4 replies.
>
> ---
> All Refs:
>
> [1] Amin Rakhsha, et al. Policy teaching via environment poisoning: Training-time adversarial attacks against reinforcement learning.
>
> [2] Vahid Behzadan, et al. Vulnerability of deep reinforcement learning to policy induction attacks.
>
> [3] Yunhan Huang, et al. Deceptive Reinforcement Learning Under Adversarial Manipulations on Cost Signals.
>
> [4] Yizhen Wang, et at. Data Poisoning Attacks against Online Learning.
>
> [5] Alexander Turner et al. Clean-Label Backdoor Attacks
>
> [6] Yuzhe Ma, et al. Policy poisoning in batch reinforcement learning and control.
>
> [7] Haoqi Zhang, et al. Value-based policy teaching with active indirect elicitation

---

> > ### Author Response · Authors · 2020-11-16
> > **[Part 4. Ambiguous Wording] We have fixed the writing as suggested by the reviewer**
> >
> > > Q9: Why is Markovian property important in 'are no longer i.i.d. due to the Markovian property'?
> >
> > A9:
> > We apologize for the misleading wording. The data samples are no longer i.i.d., and they are Markovian. We have fixed it in our manuscript.
> >
> > ---
> > > Q10: We use Atari games as a motivating example. We claim our setting doesn't assume knowledge of dynamics. However, the dynamics can be obtained in Atari games.
> >
> > A10:
> > We agree with the reviewer that we might use a better motivating example to avoid confusion. We simply meant to motivate that many RL tasks are complex, and their MDPs are large, possibly unknown. Therefore, many existing methods which compute poisoning strategy with MDP dynamic parameters[1] cannot be used. We have made the modification and used a recommender system example to motivate.
> >
> > ---
> > > Q11: The paper is motivated by Atari games, but Atari games were not used as a test-bed.
> >
> > A11:
> > As explained in A10, the motivation is that many RL environments are large-scale, continuous, even unknown, thus existing algorithms which directly compute attack strategies based on MDP parameters are not practical. In our experiments, we used multiple complex environments, such as Mujoco tasks which have continuous state space and continuous action space, and we do not assume any knowledge of the tasks. Thus our experiments match our motivation.
> > We appreciate the reviewer's suggest, and have replaced the Atari example in the introduction to avoid ambiguity.
> >
> > ---
> > > **Q12: $\psi$ and $\Psi$ not defined before Section 4.3.**
> >
> > A12:
> > We deliberately put an informal explanation of $\psi$ and $\Psi$ in the main paper as we believe their detailed definitions do not affect the description of the algorithm, and have thus deferred the formal definition to Algorithm 2 in Appendix E. However, we appreciate R#4’s suggestion and have added the explanations back to our modified paper.
> >
> >
> >
> > ---
> >
> > We greatly appreciate that R#4 points out some typos in the paper. We have already fixed them in the modified version.

---

> > ### Author Response · Authors · 2020-11-16
> > **[Part 3. Algorithm Details and Extensions] More details of the attacking process, and extension to the fully online setting**
> >
> > > **Q7: Clarification of the episodic attacking, and whether our algorithm generalizes to a fully online setting.**
> > > And **Question d** : *"Which practical application would support the episodic poisoning setting described in this paper?""*
> >
> > A7:
> > **Why episodic.** Although our algorithm is amenable to fully online setting, we would like to emphasize that our setting is realistic since most existing on-policy policy gradient deep RL methods (e.g. PPO) follow the episodic setting as we described in Section 3.2 "Procedure of Online Poisoning": the agent starts from an initial policy, uses the policy to roll out a batch of trajectories (we call them observations), then uses the data to update its policy to a new one, then rollouts trajectories again and repeat this process.
> >
> > **How one can poison.** The batch of trajectories generated by the current policy are usually stored in a temporary buffer, and the agent accesses this buffer when it updates the policy. Thus, the attacker can simply alter the buffer to poison these trajectories.
> >
> > **A practical application.** Consider a simple RL-based item recommendation system, where a user's information is a state; a recommended item is an action; a positive reward is given to the agent if the user clicks the recommended item. The agent recommends items according to its current policy; its interactions with the environment, including corresponding states, actions and rewards, are collected via the internet and possibly tapped and altered by some adversary. The states, actions and rewards stored in remote servers or local machines are also vulnerable if the hacker manipulates the stored files.
> >
> > **Our algorithm is amenable to the fully online setting.** The fully online learning where the agent updates its policy after every action usually happens for off-policy learning, since in the on-policy case, the agent needs full trajectories to evaluate its policy. And for the off-policy case, our algorithm could still work as described below:
> >
> > (a) if the adversary can manipulate the minibatch the learner samples at every step, our proposed poisoning process works as usual. (We tested it on SAC, the mean reward is 312.6 for non-poison vs 11.1 under our poison. We will report the results in our modified version.)
> >
> > (b) otherwise, if the adversary doesn’t see which minibatch the learner samples but has access to the buffer, he can still alter or insert some samples to influence learning.
> >
> >
> >
> > ---
> > >**Q8: The concrete black-box attacking process and the definition of the pseudo learner.**
> > > And **Question e**: *"How exactly is the pseudo-learner defined in this paper?"*
> >
> > A8:
> > First, the definition of pseudo learner is right in the sentence itself:
> > *A black-box attacker can train a pseudo learner by taking the same (poisoned) observations as the learner, i.e.,
> > $\tilde{\pi_k} = f (\tilde{\pi_{k-1}}, \check{\mathcal{O_{k-1}}})$*
> >
> > Here $\tilde{\pi_k}$ is a pseudo learner. $\tilde{\pi}_{k-1}$ is the pseudo learner the black-box attacker has in the last iteration. $f$ is the learner's algorithm (which implies the attacker relies on the knowledge about the learner's algorithm).
> >
> > Then, the black-box attacking process is roughly as below:
> >
> > Step 0: initialize a pseudo learner $\tilde{\pi}_0$;
> >
> > For $k = 1, 2, \cdots, K$:
> >
> > Step 1: obtains the clean data $\mathcal{O}_{k-1}$ generated by the learner's policy in current iteration by eavesdropping;
> >
> > Step 2: find a good poisoning attack by solving (P) with projected gradient descent;
> >
> > Step 3: updates the pseudo learner $\tilde{\pi_k} = f (\tilde{\pi_{k-1}}, \check{\mathcal{O_{k-1}}})$, where $\check{\mathcal{O}}_{k-1}$ is the poisoned data, and $f$ is the algorithm the learner is using;
> >
> > Step 4: send the poisoned data $\check{\mathcal{O}}_{k-1}$ to the learner.

---

> > ### Author Response · Authors · 2020-11-16
> > **[Part 2. Problem Formulation and Theory] Our setting is more general and practical than prior work; we provide thoeretical insights**
> >
> > >**Q4: Why does Problem (Q) use weighted loss?**
> >
> > A4:
> > Our goal is to formulate Problem (Q) to be general and to cover different scenarios. As explained in the paragraph under (Q), $\lambda$'s control how much the attacker values the results of different iterations. The weighted loss **makes this formulation more universally applicable to various attacking scenarios**. For example, an attacker may want to ruin the latter iterations more than the former iterations, then he can put more weights to the latter iterations.
> >
> > In Section 4, we propose a concrete algorithm for solving (Q) and analyze it. For notation briefly and readability, we set all weights to be 1. But a simple modification of our algorithm will apply to the weighted case.
> >
> >
> > > **Q5: How the problem formulation compares to prior work?**
> > > And **Question b** : *"The optimization problem (Q) seems to be different from the ones studied in (Ma et al. 2019) and (Rakhsha et al. 2020). Could you elaborate on the differences between these attack formulations? How does your setting relate to the setting of (Zhang and Parkes, 2008) in terms of computational complexity?"*
> >
> > A5:
> > We have discussed the problem formulations of other papers in our Appendix B. In the related work section, we classify papers mainly by their attacking objectives (targeted vs non-targeted) and attacker knowledge (omniscient, white-box, black-box).
> >
> > In terms of the mathematical formulation of the optimization problems, we formulate the attacker's budget and power as constraints, and the attacker's loss as objective. However, a lot of related papers[1,6] formulate the attacker's goal as a constraint (let the poisoned policy be the same with target policy), and the cost (amount of perturbation) as objectives. More differently, [3] formulates the problem as a non-constrained one, which is essentially a Lagrangian relaxation of our formulation. In different scenarios, people can choose different formulations. But our formulation is more suitable for a real-world scenario, where the attacker tries to do its best with a limited budget. Most realistic poisoning works in supervised learning have similar formulations with ours [5].
> >
> > Another important difference between our formulation and others is that we can deal with both targeted and non-targeted (reward-minimizing) poisoning, while almost all existing works only focus on targeted poisoning.
> >
> > **Comparison with Zhang and Parkes [7] in terms of computational complexity.**
> > [7] formulates the problem as a mixed integer program, and our Problem (Q) is also a mixed integer program. But the major difference is that they explicitly incorporate the dynamics (transition probability, reward function), the state values and the state-action values into the constraints of the problem. In their algorithm, they repeatedly seek solutions that satisfy the constraints and maximize the objective. The advantage of this method is that it has theoretical guarantees. However, note that the constraints in their formulation should hold for all states and actions. In large and continuous MDPs, the method in [7] becomes intractable or too expensive. In contrast, our method is more computationally efficient, although we find an approximate solution rather than a guaranteed solution. Thus, it might be better to use the method in [7] for small, known-dynamics tabular environments, and use our algorithm for large, unknown-dynamics and continuous environments.
> >
> >
> > ---
> > >**Q6: The proposed algorithm does not have theoretical guarantees.**
> >
> > A6:
> > We admit that our algorithm mainly focuses on solving the practical problem of poisoning RL with unknown dynamics in complex environments. Our poisoning algorithm works for deep RL networks and large continuous MDPs -- in these settings, it is generally hard to derive theoretical guarantees. **But we did provide some theoretical interpretations and insights about the algorithm** in Appendix F. Also, our proposed **stability radius** is a principled way of measuring the vulnerability of RL algorithms, and we show and prove a **performance guarantee for RL policy against poisoning** in Appendix D, Proposition 2.

---

> > ### Author Response · Authors · 2020-11-16
> > **[Part 1. Clarifications for Questions due to Misinterpretation]**
> >
> > > **Q1: Footnote 1 is confusing. What types of attacks are considered in this paper?**
> > > And **Question a** : *"Could you clarify what types of attacks are considered in the paper?"*
> >
> > A1:
> > R#4 might have misinterpreted the "observations" in our paper as the observed states. We would like to clarify that throughout our paper, we use "states" to denote the states (input to the agent), and the word "observations" to denote the collection of trajectories/rollouts generated by executing a policy, as explained in Section 3.2. One can regard the on-policy buffer in the implementation of deep RL as the current collection of observations.
> >
> > We also emphasize the difference between "observation" and "poison aim".
> > **Observation $\mathcal{O}$ is the collection of trajectories rolled out by the learning policy**, including sequences of states  $\mathcal{O^s}$, actions $\mathcal{O^a}$ and rewards $\mathcal{O^r}$. Footnote 1 states that we poison the observation (trajectories) rather than changing the underlying MDP as in some prior work[1].
> >
> > And we further define 3 types of "**poisoning aims**": states, actions, and rewards. For example, if the poisoning aim is the states, then the attacker only perturbs the states in the trajectories.
> >
> > Existing works usually only focus on one of these 3 poisoning aims, e.g., [2] only poisons states and [3] only poisons rewards. But our method works for these 3 types universally.
> >
> > "**Hybrid aim**" means that during poisoning, the attacker can shift its poisoning aim among state, action and reward, while non-hybrid means the attacker fixes one type of poison aim and does not change. We evaluate the performance of our algorithm for all these scenarios in experiments.
> >
> >
> >
> > ---
> > > **Q2: About Problem (Q) constraint (b). Explanation about white-box attackers and black-box attackers.**
> >
> > A2:
> > The confusion R#4 has is mainly about the knowledge of white-box attackers and black-box attackers. The definition in Section 3.2 is precise, and the description in Section 2 is more intuitive. R#4's confusion might be caused by the word "model" in Section 2. So we clarify that “model” denotes the policy/parameters.
> > A detailed definition of the attacker's knowledge is given in Appendix B, and here we further clarify the concepts:
> > - white-box attacker knows the learner's learning algorithm (e.g. PPO, DQN, etc), as well as the learner's policy model $\pi$ (i.e., knows the parameters of the policy model).
> > - black-box attacker does not know the learner's policy model $\pi$ (i.e., does not know its parameters). But a black-box attacker might know which learning algorithm the victim uses. For example, [2] studies black-box attacking, but they assume the attacker knows the victim is learning with DQN. This is why we say the black-box attacker "may know the learner’s algorithm".
> >
> >
> > ---
> > > **Q3: About Problem (Q) constraints \(c\) and (d).**
> > > And **Question c* : *"Could you explain the practical importance of imposing these two constraints together?""*
> >
> > A3:
> > As we explained in Section 3.2, \(c\) and (d) deal with two different attacking constraints. These two constraints characterize different aspects of realistic limits an attacker may have. The attack budget \(c\) restricts the attacker "to attack a limited number of iterations". The attack power (d) restricts the attacker "not to change the clean data too much". More explicitly,
> > - The attack power constraint (d) controls that the perturbed quantity should stay within a $\epsilon$-ball around the unpoisoned quantity, as widely used in various types of adversarial attacking.
> > - The attack budget constraint \(c\) is here because it is an online poisoning problem, different from the offline setting people usually focus on. As studied by another online poisoning work[4], the attacker may only be able to alter a subset of the data stream to avoid being detected.
> >
> >
> > In practice, either one of these two constraints or both can exist. We include both to provide a solution in the most general situation. But our algorithm also works if only one constraint exists.
> >
> > **R#4 might be wondering why we consider separate constraints instead of the summation of power over steps, i.e., the total power**. Our answer is that it would be more realistic to let each step have an upper limit of power, so as to lower the risk of the attacker being identified as an adversary. Think of the analogy in adversarial attacks in supervised learning, where we usually limit the amount of perturbation added to each data point (e.g. image), instead of the total perturbations to all data points.

---

> > ### Comment · AnonReviewer4 · 2020-11-17
> > **Review update**
> >
> > Thank you for providing the detailed explanations in your response. The response provides compelling arguments, and addresses most of my concerns, so I will increase my score to reflect that.

---

> > > ### Author Response · Authors · 2020-11-20
> > > **Thank you and we will further revise our paper as you suggested**
> > >
> > > Thank you for your valuable comments. We will incorporate your suggestions in our revised version and add more detailed explanations to the key points. Thanks again for your encouragement!

---

### Author Response · Authors · 2020-11-16
**Modifications to the Manuscript**

We greatly appreciate the feedback given by all reviewers. We have addressed all the questions/concerns raised by reviewers in our responses and the revised manuscript.

Poisoning RL, especially deep RL, is an emerging research field and many new notations/concepts are yet to be developed. One of our contribution is to formulate and develop such notational conventions. Therefore reviewers might have some misinterpretations of some definitions in our paper, which resulted in possible misunderstandings of the method we propose. We greatly appreciate the feedback from all reviewers that helped us improve our formulation. Now we have updated our manuscript and added some explanations/details to support our proposed problem formulation and algorithm.


### Summarization of main modifications:


1. Added Procedure 1 in Section 3.2 to illustrate the process of online learning and poisoning.

2. Added more explanation of poison aim in Section 3.3.1.

3. Modified the explanation of *(b) Imitate the Policy-Based Learner* under Problem (Q).

4. Added Algorithm 2 in Section 4.3 to illustrate the concrete attacking process, as well as detailed explanations about white-box and black-box attackers.

5. Added explanations about an extension to off-policy learners, as well as experiment results in Appendix G.4.

6. Fixed some typos and ambiguous wording.

7. Updated the remarks of Definition 1, and modified Proposition 2 in appendix.


Also, in our original submission, we put our appendix pdf in the supplementary materials, and in the rebuttal revision, we put the appendix right after the references in our main pdf to ease the reading.

If there are more questions, feel free to let us know. Thanks again for your time and all valuable comments.

---

### Decision · Program_Chairs · 2021-01-07
**Final Decision**

**Decision:**

Accept (Poster)

**Comment:**

The paper focuses on adversarial attacks for RL, which is an exciting understudied research direction, and can be of interest to the community. All the reviewers are (mildly) positive about the paper and the author competently replied to the concerns expressed by the reviewers.